# SAMPLING-FREE LEARNING OF
# BAYESIAN QUANTIZED NEURAL NETWORKS

**Jiahao Su**
Department of Electrical and Computer Engineering
University of Maryland
College Park, MD 20740
jiahaosu@umd.edu

**Milan Cvitkovic**
Amazon Web Services
Seattle, WA, USA
cvitkom@amazon.com

**Furong Huang**
Department of Computer Science
University of Maryland
College Park, MD 20740
furongh@cs.umd.edu

## ABSTRACT

Bayesian learning of model parameters in neural networks is important in scenarios where estimates with well-calibrated uncertainty are important. In this paper, we propose Bayesian quantized networks (BQNs), quantized neural networks (QNNs) for which we learn a posterior distribution over their discrete parameters. We provide a set of efficient algorithms for learning and prediction in BQNs without the need to sample from their parameters or activations, which not only allows for differentiable learning in QNNs, but also reduces the variance in gradients. We evaluate BQNs on MNIST, Fashion-MNIST, KMNIST and CIFAR10 image classification datasets. compared against bootstrap ensemble of QNNs (E-QNN). We demonstrate BQNs achieve both lower predictive errors and better-calibrated uncertainties than E-QNN (with less than 20% of the negative log-likelihood).

## 1 INTRODUCTION

A Bayesian approach to deep learning considers the network's parameters to be random variables and seeks to infer their posterior distribution given the training data. Models trained this way, called *Bayesian neural networks* (BNNs) (Wang & Yeung, 2016), in principle have well-calibrated uncertainties when they make predictions, which is important in scenarios such as active learning and reinforcement learning (Gal, 2016). Furthermore, the posterior distribution over the model parameters provides valuable information for evaluation and compression of neural networks.

There are three main challenges in using BNNs: **(1) Intractable posterior:** Computing and storing the exact posterior distribution over the network weights is intractable due to the complexity and high-dimensionality of deep networks. **(2) Prediction:** Performing a forward pass (a.k.a. as *probabilistic propagation*) in a BNN to compute a prediction for an input cannot be performed exactly, since the distribution of hidden activations at each layer is intractable to compute. **(3) Learning:** The classic *evidence lower bound* (ELBO) learning objective for training BNNs is not amenable to backpropagation as the ELBO is not an explicit function of the output of probabilistic propagation.

These challenges are typically addressed either by making simplifying assumptions about the distributions of the parameters and activations, or by using sampling-based approaches, which are expensive and unreliable (likely to overestimate the uncertainties in predictions). Our goal is to propose a **sampling-free** method which uses probabilistic propagation to deterministically learn BNNs.

A seemingly unrelated area of deep learning research is that of *quantized neural networks* (QNNs), which offer advantages of computational and memory efficiency compared to continuous-valued models. QNNs, like BNNs, face challenges in training, though for different reasons: **(4.1)** The non-

differentiable activation function is not amenable to backpropagation. **(4.2)** Gradient updates cease to be meaningful, since the model parameters in QNNs are coarsely quantized.

In this work, we combine the ideas of BNNs and QNNs in a novel way that addresses the aforementioned challenges **(1)(2)(3)(4)** in training both models. We propose *Bayesian quantized networks* (BQNs), models that (like QNNs) have quantized parameters and activations over which they learn (like BNNs) categorical posterior distributions. BQNs have several appealing properties:

- BQNs solve challenge **(1)** due to their use of categorical distributions for their model parameters.
- BQNs can be trained via sampling-free backpropagation and stochastic gradient ascent of a differentiable lower bound to ELBO, which addresses challenges **(2)**, **(3)** and **(4)** above.
- BQNs leverage efficient tensor operations for probabilistic propagation, further addressing challenge **(2)**. We show the equivalence between probabilistic propagation in BQNs and tensor contractions (Kolda & Bader, 2009), and introduce a rank-1 CP tensor decomposition (mean-field approximation) that speeds up the forward pass in BQNs.
- BQNs provide a tunable trade-off between computational resource and model complexity: using a refined quantization allows for more complex distribution at the cost of more computation.
- Sampling from a learned BQN provides an alternative way to obtain deterministic QNNs .

In our experiments, we demonstrate the expressive power of BQNs. We show that BQNs trained using our sampling-free method have much better-calibrated uncertainty compared with the state-of-the-art *Bootstrap ensemble of quantized neural networks* (E-QNN) trained by Courbariaux et al. (2016). More impressively, our trained BQNs achieve comparable log-likelihood against Gaussian *Bayesian neural network* (BNN) trained with *stochastic gradient variational Bayes* (SGVB) (Shridhar et al., 2019) (the performance of Gaussian BNNs are expected to be better than BQNs since they allows for continuous random variables). We further verify that BQNs can be easily used to compress (Bayesian) neural networks and obtain determinstic QNNs. Finally, we evaluate the effect of mean-field approximation in BQN, by comparing with its Monte-Carlo realizations, where no approximation is used. We show that our sampling-free probabilistic propagation achieves similar accuracy and log-likelihood — justifying the use of mean-field approximation in BQNs.

**Related Works.** In Appendix A, we survey different approaches for training **Bayesian neural networks** including *sampling-free assumed density filtering* (Minka, 2001; Soudry et al., 2014; Hernández-Lobato & Adams, 2015; Ghosh et al., 2016), *sampling-based variational inference* (Graves, 2011; Blundell et al., 2015; Shridhar et al., 2019), as well as *sampling-free variational inference* (Wu et al., 2018), **probabilistic neural networks** (Wang et al., 2016; Shekhovtsov & Flach, 2018; Gast & Roth, 2018), **quantized neural network** (Han et al., 2015; Courbariaux et al., 2015; Zhu et al., 2016; Kim & Smaragdis, 2016; Zhou et al., 2016; Rastegari et al., 2016; Hubara et al., 2017; Esser et al., 2015; Peters & Welling, 2018; Shayer et al., 2017), and **tensor networks and tensorial neural networks** (Grasedyck et al., 2013; Orús, 2014; Cichocki et al., 2016; 2017; Su et al., 2018; Newman et al., 2018; Robeva & Seigal, 2017).

**Contributions:**

- We propose an alternative *evidence lower bound (ELBO)* for Bayesian neural networks such that optimization of the variational objective is compatible with the backpropagation algorithm.
- We introduce Bayesian quantized networks (BQNs), establish a duality between BQNs and hierarchical tensor networks, and show prediction a BQN is equivalent to a series of tensor contractions.
- We derive a sampling-free approach for both learning and inference in BQNs using probabilistic propagation (analytical inference), achieving better-calibrated uncertainty for the learned models.
- We develop a set of fast algorithms to enable efficient learning and prediction for BQNs.

## 2 BAYESIAN NEURAL NETWORKS

**Notation.** We use bold letters such as $\boldsymbol{\theta}$ to denote random variables, and non-bold letters such as $\theta$ to denote their realizations. We abbreviate $\mathbf{Pr}[\boldsymbol{\theta} = \theta]$ as $\mathbf{Pr}[\theta]$ and use bold letters in an equation if the equality holds for *arbitrary* realizations. For example, $\mathbf{Pr}[\boldsymbol{x}, \boldsymbol{y}] = \mathbf{Pr}[\boldsymbol{y}|\boldsymbol{x}]\,\mathbf{Pr}[\boldsymbol{x}]$ means $\mathbf{Pr}[\boldsymbol{x} = x, \boldsymbol{y} = y] = \mathbf{Pr}[\boldsymbol{y} = y|\boldsymbol{x} = x]\,\mathbf{Pr}[\boldsymbol{x} = x], \forall x \in \mathcal{X}, y \in \mathcal{Y}$.

## 2.1 PROBLEM SETTING

Given a dataset $\mathcal{D} = \{(x_n, y_n)\}_{n=1}^{N}$ of $N$ data points, we aim to learn a neural network with model parameters $\boldsymbol{\theta}$ that predict the output $y \in \mathcal{Y}$ based on the input $x \in \mathcal{X}$. **(1)** We first solve the **learning problem** to find an *approximate posterior distribution* $Q(\boldsymbol{\theta}; \phi)$ over $\boldsymbol{\theta}$ with parameters $\phi$ such that $Q(\boldsymbol{\theta}; \phi) \approx \mathbf{Pr}[\boldsymbol{\theta}|\mathcal{D}]$. **(2)** We then solve the **prediction problem** to compute the *predictive distribution* $\mathbf{Pr}[\boldsymbol{y}|x, \mathcal{D}]$ for arbitrary input $\boldsymbol{x} = x$ given $Q(\boldsymbol{\theta}; \phi)$. For notational simplicity, we will omit the conditioning on $\mathcal{D}$ and write $\mathbf{Pr}[\boldsymbol{y}|x, \mathcal{D}]$ as $\mathbf{Pr}[\boldsymbol{y}|x]$ in what follows.

In order to address the prediction and learning problems in BNNs, we analyze these models in their general form of *probabilistic graphical models* (shown in Figure 3b in Appendix B). Let $\boldsymbol{h}^{(l)}$, $\boldsymbol{\theta}^{(l)}$ and $\boldsymbol{h}^{(l+1)}$ denote the inputs, model parameters, and (hidden) outputs of the $l$-th layer respectively. We assume that $\boldsymbol{\theta}^{(l)}$'s are *layer-wise independent*, i.e. $Q(\boldsymbol{\theta}; \phi) = \prod_{l=0}^{L-1} Q(\boldsymbol{\theta}^{(l)}; \phi^{(l)})$, and $\boldsymbol{h}^{(l)}$ follow the *Markovian property*, i.e. $\mathbf{Pr}[\boldsymbol{h}^{(l+1)}|\boldsymbol{h}^{(:l)}, \boldsymbol{\theta}^{(:l)}] = \mathbf{Pr}[\boldsymbol{h}^{(l+1)}|\boldsymbol{h}^{(l)}, \boldsymbol{\theta}^{(l)}]$.

## 2.2 THE PREDICTION PROBLEM

Computing the predictive distribution $\mathbf{Pr}[\boldsymbol{y}|x, \mathcal{D}]$ with a BNN requires marginalizing over the random variable $\boldsymbol{\theta}$. The hierarchical structure of BNNs allows this marginalization to be performed in multiple steps sequentially. In Appendix B, we show that the predictive distribution of $\boldsymbol{h}^{(l+1)}$ given input $\boldsymbol{x} = x$ can be obtained from its preceding layer $\boldsymbol{h}^{(l)}$ by

$$\underbrace{\mathbf{Pr}[\boldsymbol{h}^{(l+1)}|x]}_{P(\boldsymbol{h}^{(l+1)}; \psi^{(l+1)})} = \int_{h^{(l)}, \theta^{(l)}} \mathbf{Pr}[\boldsymbol{h}^{(l+1)}|h^{(l)}, \theta^{(l)}] \, Q(\theta^{(l)}; \phi^{(l)}) \underbrace{\mathbf{Pr}[h^{(l)}|x]}_{P(h^{(l)}; \psi^{(l)})} dh^{(l)} d\theta^{(l)} \tag{1}$$

This iterative process to compute the predictive distributions layer-by-layer sequentially is known as *probabilistic propagation* (Soudry et al., 2014; Hernández-Lobato & Adams, 2015; Ghosh et al., 2016). With this approach, we need to explicitly compute and store each intermediate result $\mathbf{Pr}[\boldsymbol{h}^{(l)}|x]$ in its parameterized form $P(\boldsymbol{h}^{(l)}; \psi^{(l)})$ (the conditioning on $x$ is hidden in $\psi^{(l)}$, i.e. $\psi^{(l)}$ is a function of $x$). Therefore, probabilistic propagation is a deterministic process that computes $\psi^{(l+1)}$ as a function of $\psi^{(l)}$ and $\phi^{(l)}$, which we denote as $\psi^{(l+1)} = g^{(l)}(\psi^{(l)}, \phi^{(l)})$.

**Challenge in Sampling-Free Probabilistic Propagation.** If the hidden variables $\boldsymbol{h}^{(l)}$'s are continuous, Equation (1) generally can not be evaluated in closed form as it is difficult to find a family of parameterized distributions $P$ for $\boldsymbol{h}^{(l)}$ such that $\boldsymbol{h}^{(l+1)}$ remains in $P$ under the operations of a neural network layer. Therefore most existing methods consider approximations at each layer of probabilistic propagation. In Section 4, we will show that this issue can be (partly) addressed if we consider the $\boldsymbol{h}^{(l)}$'s to be discrete random variables, as in a BQN.

## 2.3 THE LEARNING PROBLEM

**Objective Function.** A standard approach to finding a good approximation $Q(\boldsymbol{\theta}; \phi)$ is *variational inference*, which finds $\phi^{\star}$ such that the *KL-divergence* $\mathbf{KL}(Q(\boldsymbol{\theta}; \phi)||\mathbf{Pr}[\boldsymbol{\theta}|\mathcal{D}])$ from $Q(\boldsymbol{\theta}; \phi)$ to $\mathbf{Pr}[\boldsymbol{\theta}|\mathcal{D}]$ is minimized. In Appendix B, we prove that to minimizing the KL-divergence is equivalent to maximizing an objective function known as the *evidence lower bound (ELBO)*, denoted as $\mathcal{L}(\phi)$.

$$\max_{\phi} \mathcal{L}(\phi) = -\mathbf{KL}(Q(\boldsymbol{\theta}; \phi)||\mathbf{Pr}[\boldsymbol{\theta}|\mathcal{D}]) = \sum_{n=1}^{N} \mathcal{L}_n(\phi) + \mathcal{R}(\phi) \tag{2}$$

$$\text{where } \mathcal{L}_n(\phi) = \mathbb{E}_Q \left[\log \mathbf{Pr}[y_n|x_n, \boldsymbol{\theta}]\right] \text{ and } \mathcal{R}(\phi) = \mathbb{E}_Q \left[\log \left(\mathbf{Pr}[\boldsymbol{\theta}]\right)\right] + H(Q)$$

**Probabilistic Backpropagation.** Optimization in neural networks heavily relies on the gradient-based methods, where the partial derivatives $\partial \mathcal{L}(\phi)/\partial \phi$ of the objective $\mathcal{L}(\phi)$ w.r.t. the parameters $\phi$ are obtained by *backpropagation*. Formally, if the output produced by a neural network is given by a (sub-)differentiable function $g(\phi)$, and the objective $\mathcal{L}(g(\phi))$ is an *explicit* function of $g(\phi)$ (and not just an explicit function of $\phi$), then the partial derivatives can be computed by chain rule:

$$\partial \mathcal{L}(g(\phi))/\partial \phi = \partial \mathcal{L}(g(\phi))/\partial g(\phi) \cdot \partial g(\phi)/\partial \phi \tag{3}$$

The learning problem can then be (approximately) solved by first-order methods, typically *stochastic gradient descent/ascent*. Notice that **(1)** For classification, the function $g(\phi)$ returns the probabilities after the softmax function, not the categorical label; **(2)** An additional regularizer $\mathcal{R}(\phi)$ on the parameters will not cause difficulty in backpropagation, given $\partial\mathcal{R}(\phi)/\partial\phi$ is easily computed.

**Challenge in Sampling-Free Probabilistic Backpropagation.** Learning BNNs is not amenable to standard backpropagation because the ELBO objective function $\mathcal{L}(\phi)$ in (4b) is not an explicit (i.e. *implicit*) function of the predictive distribution $g(\phi)$ in (4a):

$$g_n(\phi) = \mathbb{E}_Q\left[\mathbf{Pr}[y_n|x_n,\boldsymbol{\theta}]\right] = \int_\theta \mathbf{Pr}[y_n|x_n,\theta]Q(\theta;\phi)d\theta \tag{4a}$$

$$\mathcal{L}_n(\phi) = \mathbb{E}_Q\left[\log(\mathbf{Pr}[y_n|x_n,\boldsymbol{\theta}])\right] = \int_\theta \log\left(\mathbf{Pr}[y_n|x_n,\theta]\right)Q(\theta;\phi)d\theta \tag{4b}$$

Although $\mathcal{L}_n(\phi)$ is a function of $\phi$, it is not an explicit function of $g_n(\phi)$. Consequently, the chain rule in Equation (3) on which backpropagation is based is not directly applicable.

# 3 PROPOSED LEARNING METHOD FOR BAYESIAN NEURAL NETWORKS

**Alternative Evidence Lower Bound.** We make learning in BNNs amenable to backpropagation by developing a lower bound $\overline{\mathcal{L}}_n(\phi) \leq \mathcal{L}_n(\phi)$ such that $\partial\overline{\mathcal{L}}_n(\phi)/\partial\phi$ can be obtained by chain rule (i.e. $\overline{\mathcal{L}}_n(\phi)$ is an explicit function of the results from the forward pass.) With $\overline{\mathcal{L}}_n(\phi)$ in hand, we can (approximately) find $\phi^\star$ by maximizing the alternative objective via gradient-based method:

$$\phi^\star = \arg\max_\phi \overline{\mathcal{L}}(\phi) = \arg\max_\phi \left(\mathcal{R}(\phi) + \sum_{n=1}^N \overline{\mathcal{L}}_n(\phi)\right) \tag{5}$$

In Appendix C.1, we proved one feasible $\overline{\mathcal{L}}_n(\phi)$ which only depends on second last output $\boldsymbol{h}^{(L-1)}$.

**Theorem 3.1 (Alternative Evidence Lower Bound).** *Define each term $\overline{\mathcal{L}}_n(\phi)$ in $\overline{\mathcal{L}}(\phi)$ as*

$$\overline{\mathcal{L}}_n(\phi) := \mathbb{E}_{\boldsymbol{h}^{(L-1)}\sim P;\ \boldsymbol{\theta}^{(L-1)}\sim Q}\left[\log\left(\mathbf{Pr}[y_n|\boldsymbol{h}^{(L-1)},\boldsymbol{\theta}^{(L-1)}]\right)\right] \tag{6}$$

*then $\overline{\mathcal{L}}_n(\phi)$ is a lower bound of $\mathcal{L}_n(\phi)$, i.e. $\overline{\mathcal{L}}_n(\phi) \leq \mathcal{L}_n(\phi)$. The equality $\overline{\mathcal{L}}_n(\phi) = \mathcal{L}_n(\phi)$ holds if $\boldsymbol{h}^{(L-1)}$ is deterministic given input $x$ and all parameters before the last layer $\theta^{(:\,L-2)}$.*

**Analytic Forms of $\overline{\mathcal{L}}_n(\phi)$.** While the lower bound in Theorem 3.1 applies to BNNs with *arbitrary* distributions $P$ on hidden variables $\boldsymbol{h}$, $Q$ on model parameters $\boldsymbol{\theta}$, and *any* problem setting (e.g. classification or regression), *in practice* sampling-free probabilistic backpropagation requires that $\overline{\mathcal{L}}_n(\phi)$ can be analytically evaluated (or further lower bounded) in terms of $\phi^{(L-1)}$ and $\theta^{(L-1)}$. This task is nontrivial since it requires redesign of the output layer, i.e. the function of $\mathbf{Pr}[y|\boldsymbol{h}^{(L-1)},\boldsymbol{\theta}^{(L-1)}]$. In this paper, we develop two layers for classification and regression tasks, and present the classification case in this section due to space limit. Since $\overline{\mathcal{L}}_n(\phi)$ involves the last layer only, we omit the superscripts/subscripts of $\boldsymbol{h}^{(L-1)}$, $\psi^{(L-1)}$, $\phi^{(L-1)}$, $x_n$, $y_n$, and denote them as $\boldsymbol{h}$, $\psi$, $\phi$, $x$, $y$ .

**Theorem 3.2 (Analytic Form of $\overline{\mathcal{L}}_n(\phi)$ for Classification).** *Let $\boldsymbol{h} \in \mathbb{R}^K$ (with $K$ the number of classes) be the pre-activations of a softmax layer (a.k.a. logits), and $\phi = s \in \mathbb{R}^+$ be a scaling factor that adjusts its scale such that $\mathbf{Pr}[y = c|\boldsymbol{h},s] = \exp(\boldsymbol{h}_c/s)/\sum_{k=1}^K \exp(\boldsymbol{h}_k/s)$. Suppose the logits $\{\boldsymbol{h}_k\}_{k=1}^K$ are pairwise independent (which holds under mean-field approximation) and $\boldsymbol{h}_k$ follows a Gaussian distribution $\boldsymbol{h}_k \sim \mathcal{N}(\mu_k,\nu_k)$ (therefore $\psi = \{\mu_k,\nu_k\}_{k=1}^K$) and $s$ is a deterministic parameter. Then $\overline{\mathcal{L}}_n(\phi)$ is further lower bounded as $\overline{\mathcal{L}}_n(\phi) \geq \frac{\mu_c}{s} - \log\left(\sum_{k=1}^K \exp\left(\frac{\mu_k}{s} + \frac{\nu_k}{2s^2}\right)\right)$.*

The regression case and proofs for both layers are deferred to Appendix C.

# 4 BAYESIAN QUANTIZED NETWORKS (BQNS)

While Section 3 provides a general solution to learning in BNNs, the solution relies on the ability to perform probabilistic propagation efficiently. To address this, we introduce Bayesian quantized

networks (BQNs) — BNNs where both hidden units $\boldsymbol{h}^{(l)}$'s and model parameters $\boldsymbol{\theta}^{(l)}$'s take discrete values — along with a set of novel algorithms for efficient sampling-free probabilistic propagation in BQNs. For simplicity of exposition, we assume activations and model parameters take values from the same set $\mathbb{Q}$, and denote the *degree of quantization* as $D = |\mathbb{Q}|$, (e.g. $\mathbb{Q} = \{-1, 1\}$, $D = 2$).

## 4.1 PROBABILISTIC PROPAGATION AS TENSOR CONTRACTIONS

**Lemma 4.1** (**Probabilistic Propagation in BQNs**). *After quantization, the iterative step of probabilistic propagation in Equation* (1) *is computed with a finite sum instead of an integral:*

$$P(\boldsymbol{h}^{(l+1)}; \psi^{(l+1)}) = \sum_{h^{(l)}, \theta^{(l)}} \mathbf{Pr}[\boldsymbol{h}^{(l+1)}|h^{(l)}, \theta^{(l)}] \, Q(\theta^{(l)}; \phi^{(l)}) \, P(h^{(l)}; \psi^{(l)}) \qquad (7)$$

*and a categorically distributed $\boldsymbol{h}^{(l)}$ results in $\boldsymbol{h}^{(l+1)}$ being categorical as well. The equation holds without any assumption on the operation $\mathbf{Pr}[\boldsymbol{h}^{(l+1)}|\boldsymbol{h}^{(l)}, \theta^{(l)}]$ performed in the neural network.*

Notice all distributions in Equation (7) are represented in high-order tensors: Suppose there are $I$ input units, $J$ output units, and $K$ model parameters at the $l$-th layer, then $\boldsymbol{h}^{(l)} \in \mathbb{Q}^I$, $\boldsymbol{\theta}^{(l)} \in \mathbb{Q}^K$, and $\boldsymbol{h}^{(l+1)} \in \mathbb{Q}^J$, and their distributions are characterized by $P(\boldsymbol{h}^{(l)}; \psi^{(l)}) \in \mathbb{R}^{D^I}$, $Q(\boldsymbol{\theta}^{(l)}; \phi^{(l)}) \in \mathbb{R}^{D^K}$, $P(\boldsymbol{h}^{(l+1)}; \psi^{(l+1)}) \in \mathbb{R}^{D^J}$, and $\mathbf{Pr}[\boldsymbol{h}^{(l+1)}|\boldsymbol{h}^{(l)}, \theta^{(l)}] \in \mathbb{R}^{D^J \times D^I \times D^K}$ respectively. Therefore, each step in probabilistic propagation is a *tensor contraction* of three tensors, which establishes the duality between BQNs and hierarchical tensor networks (Robeva & Seigal, 2017).

Since tensor contractions are differentiable w.r.t. all inputs, BQNs thus circumvent the difficulties in training QNNs (Courbariaux et al., 2015; Rastegari et al., 2016), whose outputs are not differentiable w.r.t. the discrete parameters. This result is not surprising: if we consider learning in QNNs as an *integer programming (IP)* problem, solving its Bayesian counterpart is equivalent to the approach to relaxing the problem into a *continuous optimization* problem (Williamson & Shmoys, 2011).

**Complexity of Exact Propagation.** The computational complexity to evaluate Equation (7) is exponential in the number of random variables $O(D^{IJK})$, which is intractable for quantized neural network of any reasonable size. We thus turn to approximations.

## 4.2 APPROXIMATE PROPAGATION VIA RANK-1 TENSOR CP DECOMPOSITION

We propose a principled approximation to reduce the computational complexity in probabilistic propagation in BQNs using *tensor CP decomposition*, which factors an intractable high-order probability tensor into tractable lower-order factors (Grasedyck et al., 2013). In this paper, we consider the simplest *rank-1 tensor CP decomposition*, where the joint distributions of $P$ and $Q$ are fully factorized into products of their marginal distributions, thus equivalent to the *mean-field approximation* (Wainwright et al., 2008). With rank-1 CP decomposition on $P(h^{(l)}; \psi^{(l)}), \forall l \in [L]$, the tensor contraction in (7) reduces to a standard *Tucker contraction* (Kolda & Bader, 2009)

$$P(\boldsymbol{h}_j^{(l+1)}; \psi_j^{(l+1)}) \approx \sum_{h^{(l)}, \theta^{(l)}} \mathbf{Pr}[\boldsymbol{h}_j^{(l+1)}|\theta^{(l)}, h^{(l)}] \prod_k Q(\theta_k^{(l)}; \phi_k^{(l)}) \prod_i P(h_i^{(l)}; \psi_i^{(l)}) \qquad (8)$$

where each term of $\psi_i^{(l)}, \phi_k^{(l)}$ parameterizes a single categorical variable. In our implementation, we store the parameters in their log-space, i.e. $Q(\theta_k^{(l)} = \mathbb{Q}(d)) = \exp(\psi_k^{(l)}(d))/\sum_{q=1}^{D} \exp(\phi_k^{(l)}(q))$.

**Fan-in Number $E$.** In a practical model, for the $l$-th layer, an output unit $\boldsymbol{h}_j^{(l+1)}$ only (conditionally) depends on a subset of all input units $\{\boldsymbol{h}_i^{(l)}\}$ and model parameters $\{\boldsymbol{\theta}_k^{(h)}\}$ according to the connectivity pattern in the layer. We denote the set of dependent input units and parameters for $\boldsymbol{h}_j^{(l+1)}$ as $\mathcal{I}_j^{(l+1)}$ and $\mathcal{M}_j^{(l+1)}$, and define the *fan-in number $E$* for the layer as $\max_j \left|\mathcal{I}_j^{(l+1)}\right| + \left|\mathcal{M}_j^{(l+1)}\right|$.

**Complexity of Approximate Propagation.** The approximate propagation reduces the computational complexity from $O(D^{IJK})$ to $O(JD^E)$, which is linear in the number of output units $J$ if we assume the fan-in number $E$ to be a constant (i.e. $E$ is not proportional to $I$).

### 4.3 FAST ALGORITHMS FOR APPROXIMATE PROPAGATION

Different types of network layers have different fan-in numbers $E$, and for those layers with $E$ greater than a small constant, Equation (8) is inefficient since the complexity grows exponential in $E$. Therefore in this part, we devise fast(er) algorithms to further lower the complexity.

**Small Fan-in Layers: Direct Tensor Contraction.** If $E$ is small, we implement the approximate propagation through *tensor contraction* in Equation (8). The computational complexity is $O(JD^E)$ as discussed previously. See Appendix D.1 for a detailed discussion.

**Medium Fan-in Layers: Discrete Fourier Transform.** If $E$ is medium, we implement approximate propagation through *fast Fourier transform* since summation of discrete random variables is equivalent to convolution between their probability mass function. See Appendix D.2 for details. With the *fast Fourier transform*, the computational complexity is reduced to $O(JE^2 D \log(ED))$.

**Large Fan-in Layers: Lyapunov Central Limit Theorem.** In a typical linear layer, the fan-in $E$ is large, and a super-quadratic algorithm using fast Fourier transform is still computational expensive. Therefore, we derive a faster algorithm based on the *Lyapunov central limit theorem* (See App D.3) With CLT, the computational complexity is further reduced to $O(JED)$.

*Remarks:* Depending on the fan-in numbers $E$, we adopt CLT for linear layers with sufficiently large $E$ such as *fully connected layers* and *convolutional layers*; DFT for those with medium $E$ such as *average pooling layers* and *depth-wise layers*; and direct tensor contraction for those with small $E$ such as *shortcut layers* and *nonlinear layers*.

## 5 EXPERIMENTS

In this section, we demonstrate the effectiveness of BQNs on the MNIST, Fashion-MNIST, KM-NIST and CIFAR10 classification datasets. We evaluate our BQNs with both *multi-layer perceptron* (MLP) and *convolutional neural network* (CNN) models. In training, each image is augmented by a random shift within 2 pixels (with an additional random flipping for CIFAR10), and no augmentation is used in test. In the experiments, we consider a class of quantized neural networks, with both *binary* weights and activations (i.e. $\mathbb{Q} = \{-1, 1\}$) with sign activations $\sigma(\cdot) = \text{sign}(\cdot)$. For BQNs, the distribution parameters $\phi$ are initialized by Xavier's uniform initializer, and all models are trained by ADAM optimizer (Kingma & Ba, 2014) for 100 epochs (and 300 epochs for CIFAR10) with batch size 100 and initial learning rate $10^{-2}$, which decays by 0.98 per epoch.

| Methods | MNIST | | KMNIST | | Fashion-MNIST | | CIFAR10 | |
|---|---|---|---|---|---|---|---|---|
| | NLL ($10^{-3}$) | % Err. | NLL ($10^{-3}$) | % Err. | NLL ($10^{-3}$) | % Err. | NLL ($10^{-3}$) | % Err. |
| E-QNN on MLP | 546.6±157.9 | 3.30 ±0.65 | 2385.6±432.3 | 17.88±1.86 | 2529.4±276.7 | 13.02±0.81 | N/A | N/A |
| BQN on MLP | **130.0±3.5** | **2.49±0.08** | **457.7±13.8** | **13.41±0.12** | **417.3±8.1** | **9.99±0.20** | N/A | N/A |
| E-QNN on CNN | 425.3±61.8 | 0.85±0.13 | 3755.7±465.1 | 11.49±1.16 | 1610.7±158.4 | **3.02±0.37** | 7989.7 ± 600.2 | 15.92 ± 0.72 |
| BQN on CNN | **41.8±1.6** | **0.85±0.06** | **295.5±1.4** | **9.95±0.15** | **209.5±2.8** | 4.65±0.15 | **530.6 ± 23.0** | **13.74 ±0.47** |

**Table 1:** Comparison of performance of BQNs against the baseline E-QNN. Each E-QNN is an ensemble of 10 networks, which are trained individually and but make predictions jointly. We report both NLL (which accounts for prediction uncertainty) and 0-1 test error (which doesn't account for prediction uncertainty). All the numbers are averages over 10 runs with different seeds, the standard deviation are exhibited following the ± sign.

**Training Objective of BQNs.** To allow for customized level of uncertainty in the learned Bayesian models, we introduce a regularization coefficient $\lambda$ in the alternative ELBO proposed in Equation (5) (i.e. a lower bound of the likelihood), and train the BQNs by maximizing the following objective:

$$\overline{\mathcal{L}}(\phi) = \sum_{n=1}^{N} \overline{\mathcal{L}}_n(\phi) + \lambda \mathcal{R}(\phi) = \lambda \left( 1/\lambda \sum_{n=1}^{N} \overline{\mathcal{L}}_n(\phi) + \mathcal{R}(\phi) \right) \tag{9}$$

where $\lambda$ controls the uncertainty level, i.e. the importance weight of the prior over the training set.

**Baselines. (1)** We compare our BQN against the baseline – *Bootstrap ensemble of quantized neural networks* (E-QNN). Each member in the ensemble is trained in a non-Bayesian way (Courbariaux et al., 2016), and jointly make the prediction by averaging over the logits from all members. Note

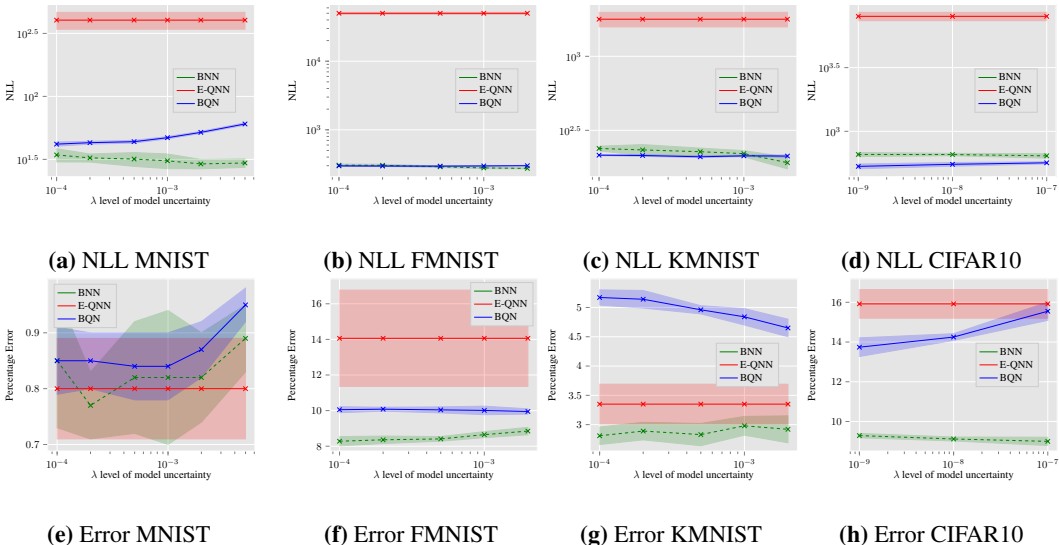

**Figure 1:** Comparison of the predictive performance of our BQNs against the E-QNN as well as the non-quantized BNN trained by SGVB on a CNN. Negative log-likelihood (NLL) which accounts for uncertainty and 0-1 test error which doesn't account for uncertainty are displayed.

that Courbariaux et al. (2016) is chosen over other QNN training methods as the baseline since it trains QNN from random initialization, thus a fair comparison to our approach. Details are discussed in Appendix A. (**2**) To exhibit the effectiveness of our BQN, we further compare against *continuous-valued Bayesian neural network* (abbreviated as BNN) with Gaussian parameters. The model is trained with *stochastic gradient variational Bayes* (SGVB) augmented by *local re-parameterization trick* (Shridhar et al., 2019). Since the BNN allows for continuous parameters (different from BQN with quantized parameters), the predictive error is expected to be lower than BQN.

**Evaluation of BQNs.** While *0-1 test error* is a popular metric to measure the predictive performance, it is too coarse a metric to assess the uncertainty in decision making (for example it does not account for how badly the wrong predictions are). Therefore, we will mainly use the *negative log-likelihood* (NLL) to measure the predictive performance in the experiments.

Once a BQN is trained (i.e. an approximate posterior $Q(\boldsymbol{\theta})$ is learned), we consider three modes to evaluate the behavior of the model: (**1**) *analytic inference (AI)*, (**2**) *Monte Carlo (MC) sampling* and (**3**) *Maximum a Posterior (MAP) estimation*:

1. In analytic inference (AI, i.e. our proposed method), we analytically integrate over $Q(\boldsymbol{\theta})$ to obtain the predictive distribution as in the training phase. Notice that the exact NLL is not accessible with probabilistic propagation (which is why we propose an alternative ELBO in Equation (5)), we will report *an upper bound* of the NLL in this mode.

2. In MC sampling, $S$ sets of model parameters are drawn independently from the posterior posterior $\theta_s \sim Q(\boldsymbol{\theta}), \forall s \in [S]$, and the forward propagation is performed as in (non-Bayesian) quantized neural network for each set $\theta_s$, followed by an average over the model outputs. The difference between analytic inference and MC sampling will be used to evaluate (**a**) the effect of mean-field approximation and (**b**) the tightness of the our proposed alternative ELBO.

3. MAP estimation is similar to MC sampling, except that only one set of model parameters $\theta^\star$ is obtained $\theta^\star = \arg\max_\theta Q(\theta)$. We will exhibit our model's ability to compress a Bayesian neural network by comparing MAP estimation of our BQN with non-Bayesian QNN.

### 5.1 Analysis of Results

**Expressive Power and Uncertainty Calibration in BQNs.** We report the performance via all evaluations of our BQN models against the Ensemble-QNN in Table 1 and Figure 1. (**1**) Compared to E-QNNs, our BQNs have significantly lower NLL and smaller predictive error (except for

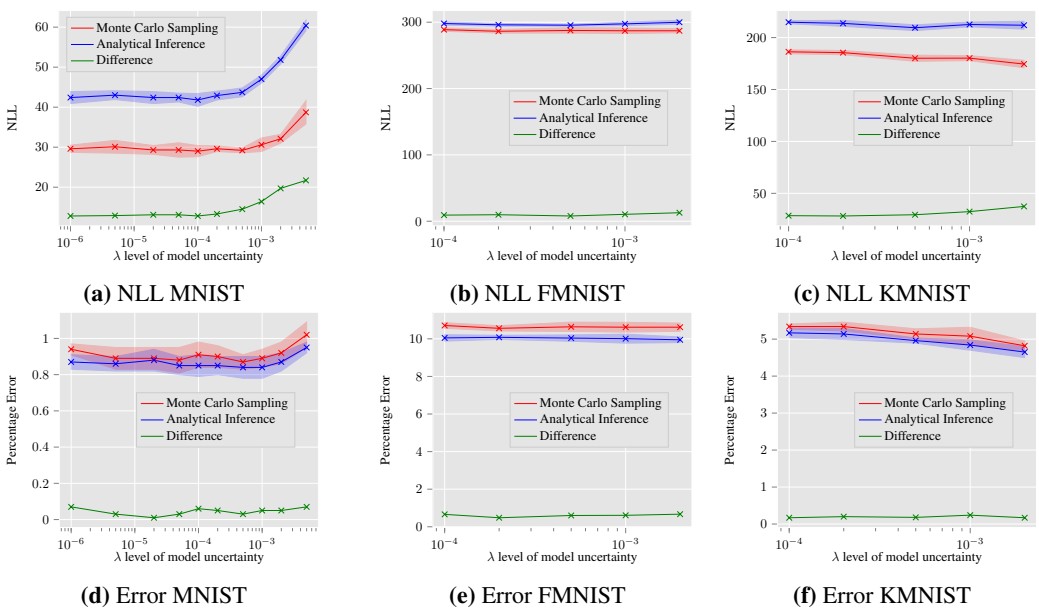

**Figure 2:** Illustration of mean-field approximation and tightness of alternative ELBO on a CNN. The performance gap between our analytical inference and the Monte Carlo Sampling is displayed.

Fashion-MNIST with architecture CNN). **(2)** As we can observe in Figure 1, BQNs impressively achieve comparable NLL to continuous-valued BNN, with slightly higher test error. As our model parameters only take values $\{-1, 1\}$, small degradation in predictive accuracy is expected.

**Evaluations of Mean-field Approximation and Tightness of the Alternative ELBO.** If analytic inference (by probabilistic propagation) were computed exactly, the evaluation metrics would have been equal to the ones with MC sampling (with infinite samples). Therefore we can evaluate the approximations in probabilistic propagation, namely *mean-field approximation* in Equation (8) and *relaxation of the original ELBO* in Equation (5), by measuring the gap between analytic inference and MC sampling. As shown in Figure 2, such gaps are small for all scenarios, which justifies the approximations we use in BQNs.

To further decouple these two factors of *mean-field approximation* and *relaxation of the original ELBO*, we vary the regularization coefficient $\lambda$ in the learning objective. **(1)** For $\lambda = 0$ (where the prior term is removed), the models are forced to become deterministic during training. Since the deterministic models do not have mean-field approximation in the forward pass, the gap between analytic inference and MC-sampling reflects the tightness of our alternative ELBO. **(2)** As $\lambda$ increases, the gaps increases slightly as well, which shows that the mean-field approximation becomes slightly less accurate with higher learned uncertainty in the model.

| Methods | MNIST | | KMNIST | | Fashion-MNIST | |
|---|---|---|---|---|---|---|
| | NLL($10^{-3}$) | % Err. | NLL($10^{-3}$) | % Err. | NLL($10^{-3}$) | % Err. |
| QNN on MLP | 522.4±42.2 | 4.14±0.25 | 2019.1±281.2 | 19.56±1.97 | 2427.1±193.5 | 15.67±1.19 |
| MAP of BQN on MLP | **137.60±4.40** | **3.69±0.09** | **464.60±12.80** | **14.79±0.21** | **461.30±13.40** | **12.89±0.17** |
| QNN on CNN | 497.4±139.5 | 1.08±0.2 | 4734.5±1697.2 | 14.2±2.29 | 1878.3±223.8 | **3.88±0.33** |
| MAP of BQN on CNN | **30.3±1.6** | **0.92±0.07** | **293.6±4.4** | **10.82±0.37** | **179.1±4.4** | 5.00±0.11 |

**Table 2:** Deterministic model compression through direct training of QNN (Courbariaux et al., 2016) v.s. MAP estimation in our proposed BQN. All the numbers are averages over 10 runs with different seeds, the standard deviation are exhibited following the ± sign.

**Compression of Neural Networks via BQNs.** One advantage of BQNs over continuous-valued BNNs is that deterministic QNNs can be obtained for free, since a BQN can be interpreted as an ensemble of infinite QNNs (each of which is a realization of posterior distribution). **(1)** One simple approach is to set the model parameters to their *MAP estimates*, which compresses a given BQN to $1/64$ of its original size (and has the same number of bits as a single QNN). **(2)** *MC sampling* can be

| Methods | MNIST | | KMNIST | | Fashion-MNIST | |
|---|---|---|---|---|---|---|
| | NLL($10^{-3}$) | % Err. | NLL($10^{-3}$) | % Err. | NLL($10^{-3}$) | % Err. |
| E-QNN on MLP | 546.60±157.90 | 3.30 ±0.65 | 2385.60±432.30 | 17.88±1.86 | 2529.40±276.70 | 13.02±0.81 |
| MC of BQN on MLP | **108.9±2.6** | **2.73±0.09** | **429.50±11.60** | **13.83±0.12** | **385.30±5.10** | **10.81±0.44** |
| E-QNN on CNN | 425.3±61.80 | **0.85±0.13** | 3755.70±465.10 | 11.49±1.16 | 1610.70±158.40 | **3.02±0.37** |
| MC of BQN on CNN | **29.2±0.6** | 0.87±0.04 | **286.3±2.7** | **10.56±0.14** | **174.5±3.6** | 4.82±0.13 |

**Table 3:** Bayesian Model compression through direct training of Ensemble-QNN vs a Monte-Carlo sampling on our proposed BQN. Each ensemble consists of 5 quantized neural networks, and for fair comparison we use 5 samples for Monte-Carlo evaluation. All the numbers are averages over 10 runs with different seeds, the standard deviation are exhibited following the $\pm$ sign.

interpreted as another approach to compress a BQN, which reduces the original size to its $S/64$ (with the same number of bits as an ensemble of $S$ QNNs). In Tables 2 and 3, we compare the models by both approaches to their counterparts (a single QNN for MAP, and E-QNN for MC sampling) trained from scratch as in Courbariaux et al. (2016). For both approaches, our compressed models outperform their counterparts (in NLL) . We attribute this to two factors: **(a)** QNNs are not trained in a Bayesian way, therefore the uncertainty is not well calibrated; and **(b)** Non-differentiable QNNs are unstable to train. Our compression approaches via BQNs simultaneously solve both problems.

## 6 CONCLUSION

We present a sampling-free, backpropagation-compatible, variational-inference-based approach for learning Bayesian quantized neural networks (BQNs). We develop a suite of algorithms for efficient inference in BQNs such that our approach scales to large problems. We evaluate our BQNs by Monte-Carlo sampling, which proves that our approach is able to learn a proper posterior distribution on QNNs. Furthermore, we show that our approach can also be used to learn (ensemble) QNNs by taking maximum a posterior (or sampling from) the posterior distribution.

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

# Appendix: Sampling-Free Learning of Bayesian Quantized Neural Networks

## A  RELATED WORK

**Probabilistic Neural Networks and Bayesian Neural Networks**   These models consider weights to be random variables and aim to learn their distributions. To further distinguish two families of such models, we call a model *Bayesian neural network* if the distributions are learned using a prior-posterior framework (i.e. via Bayesian inference) (Soudry et al., 2014; Hernández-Lobato & Adams, 2015; Ghosh et al., 2016; Graves, 2011; Blundell et al., 2015; Shridhar et al., 2019), and otherwise *probabilistic neural network* (Wang et al., 2016; Shekhovtsov & Flach, 2018; Gast & Roth, 2018). In particular, our work is closely related to *natural-parameters networks (NPN)* (Wang et al., 2016), which consider both weights and activations to be random variables from exponential family. Since categorical distribution (over quantized values) belongs to exponential family, our BQN can be interpreted as categorical NPN, but we learn the distributions via Bayesian inference.

For Bayesian neural networks, various types of approaches have been proposed to learn the posterior distribution over model parameters.

*(1) Sampling-free Assumed Density Filtering (ADF)*, including EBP (Soudry et al., 2014) and PBP (Hernández-Lobato & Adams, 2015), is an online algorithm which (approximately) updates the posterior distribution by Bayes' rule for each observation. If the model parameters $\theta$ are Gaussian distributed, Minka (2001) shows that the Bayes' rule can be computed in analytic form based on $\partial \log(g_n(\phi))/\partial\phi$, and EBP Soudry et al. (2014) derives a similar rule for Bernoulli parameters in binary classification. Notice that ADF is compatible to backpropagation:

$$\frac{\partial \log(g_n(\phi))}{\partial \phi} = \frac{1}{g_n(\phi)} \cdot \frac{\partial g_n(\phi)}{\partial \phi} \tag{10}$$

assuming $g_n(\phi)$ can be (approximately) computed by sampling-free probabilistic propagation as in Section 2. However, this approach has two major limitations: (a) the Bayes' rule needed to be derived case by case, and analytic rule for most common cases are not known yet. (b) it is not compatible to modern optimization methods (such as SGD or ADAM) as the optimization is solved analytically for each data point, therefore difficult to cope with large-scale models.

*(2) Sampling-based Variational inference (SVI)*, formulates an optimization problem and solves it approximately via *stochastic gradient descent (SGD)*. The most popular method among all is, *Stochastic Gradient Variational Bayes (SGVB)*, which approximates $\mathcal{L}_n(\phi)$ by the average of multiple samples (Graves, 2011; Blundell et al., 2015; Shridhar et al., 2019). Before each step of learning or prediction, a number of independent samples of the model parameters $\{\theta_s\}_{s=1}^S$ are drawn according to the current estimate of $Q$, i.e. $\theta_s \sim Q$, by which the predictive function $g_n(\phi)$ and the loss $\mathcal{L}_n(\phi)$ can be approximated by

$$g_n(\phi) \approx \frac{1}{S} \sum_{s=1}^S \mathbf{Pr}[y_n|x_n, \theta_s] = \frac{1}{S} \sum_{s=1}^S f_n(\theta_s) \tag{11a}$$

$$\mathcal{L}_n(\phi) \approx \frac{1}{S} \sum_{s=1}^S \log\left(\mathbf{Pr}[y_n|x_n, \theta_s]\right) = \frac{1}{S} \sum_{s=1}^S \log\left(f_n(\theta_s)\right) \tag{11b}$$

where $f_n(\theta) = \mathbf{Pr}[y_n|x_n, \theta]$ denotes the predictive function given a specific realization $\theta$ of the model parameters. The gradients of $\mathcal{L}_n(\phi)$ can now be approximated as

$$\frac{\partial \mathcal{L}_n(\phi)}{\partial \phi} \approx \frac{1}{S} \sum_{s=1}^S \frac{\partial \mathcal{L}_n(\phi)}{\partial f_n(\theta_s)} \cdot \frac{\partial f_n(\theta_s)}{\partial \theta_s} \cdot \frac{\partial \theta_s}{\partial \phi} \tag{12}$$

This approach has multiple drawbacks: (a) Repeated sampling suffers from high variance, besides being computationally expensive in both learning and prediction phases; (b) While $g_n(\phi)$ is differentiable w.r.t. $\phi$, $f_n(\theta)$ may not be differentiable w.r.t. $\theta$. One such example is quantized neural networks, whose backpropagation is approximated by *straight through estimator* (Bengio et al.,

2013). (3) The partial derivatives $\partial\theta_s/\partial\phi$ are difficult to compute with complicated *reparameterization tricks* (Maddison et al., 2016; Jang et al., 2016).

*(3) Deterministic Variational inference (DVI)* Our approach is most similar to Wu et al. (2018), which observes that if the underlying model is deterministic, i.e. $\mathbf{Pr}[\boldsymbol{h}^{(l+1)}|\boldsymbol{h}^{(l)}, \boldsymbol{\theta}^{(l)}]$ is a dirac function

$$\mathcal{L}_n(\phi) := \mathbb{E}_{\boldsymbol{h}^{(L-1)}\sim P;\ \boldsymbol{\theta}^{(L-1)}\sim Q} \left[ \log \left( \mathbf{Pr}[y_n|\boldsymbol{h}^{(L-1)}, \boldsymbol{\theta}^{(L-1)}] \right) \right] \tag{13}$$

Our approach considers a wider scope of problem settings, where the model could be stochastic, i.e. $\mathbf{Pr}[\boldsymbol{h}^{(l+1)}|\boldsymbol{h}^{(l)}, \boldsymbol{\theta}^{(l)}]$ is an arbitrary function. Furthermore, Wu et al. (2018) considers the case that all parameters $\boldsymbol{\theta}$ are Gaussian distributed, whose sampling-free probabilistic propagation requires complicated approximation (Shekhovtsov & Flach, 2018).

**Quantized Neural Networks**    These models can be categorized into two classes: (1) *Partially quantized networks*, where only weights are discretized (Han et al., 2015; Zhu et al., 2016); (2) *Fully quantized networks*, where both weights and hidden units are quantized (Courbariaux et al., 2015; Kim & Smaragdis, 2016; Zhou et al., 2016; Rastegari et al., 2016; Hubara et al., 2017). While both classes provide compact size, low-precision neural network models, fully quantized networks further enjoy fast computation provided by specialized bit-wise operations. In general, quantized neural networks are difficult to train due to their non-differentiability. Gradient descent by back-propagation is approximated by either *straight-through estimators* (Bengio et al., 2013) or *probabilistic methods* (Esser et al., 2015; Shayer et al., 2017; Peters & Welling, 2018). Unlike these papers, we focus on Bayesian learning of fully quantized networks in this paper. Optimization of quantized neural networks typically requires dedicated loss function, learning scheduling and initialization. For example, Peters & Welling (2018) considers pre-training of a continuous-valued neural network as the initialization. Since our approach considers learning from scratch (with an uniform initialization), the performance could be inferior to prior works in terms of absolute accuracy.

**Tensor Networks and Tensorial Neural Networks**    *Tensor networks* (TNs) are widely used in numerical analysis (Grasedyck et al., 2013), quantum physiscs (Orús, 2014), and recently machine learning (Cichocki et al., 2016; 2017) to model interactions among multi-dimensional random objects. Various tensorial neural networks (TNNs) (Su et al., 2018; Newman et al., 2018) have been proposed that reduce the size of neural networks by replacing the linear layers with TNs. Recently, (Robeva & Seigal, 2017) points out the duality between probabilistic graphical models (PGMs) and TNs. I.e. there exists a bijection between PGMs and TNs. Our paper advances this line of thinking by connecting hierarchical Bayesian models (e.g. Bayesian neural networks) and hierarchical TNs.

# B    SUPERVISED LEARNING WITH BAYESIAN NEURAL NETWORKS (BNNS)

The problem settings of general Bayesian model and Bayesian neural networks for supervised learning are illustrated in Figures 3a and 3b using *graphical models*.

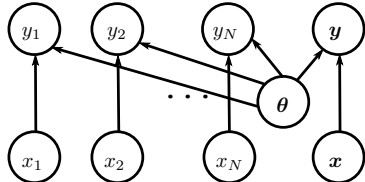

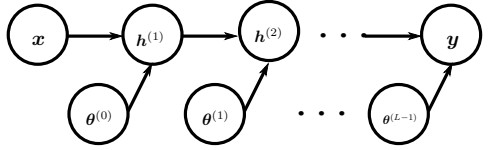

**(a)** Graphical model depiction of the problem setting in Bayesian neural networks.

**(b)** Graphical model depiction of a Bayesian neural network as a hierarchical model, where predicting $\boldsymbol{y}$ from $\boldsymbol{x}$ can be performed iteratively through the hidden variables $\boldsymbol{h}^{(l)}$'s.

**Figure 3:** Graphical models.

**General Bayesian model**    Formally, the graphical model in Figure 3a implies the joint distribution of the model parameters $\boldsymbol{\theta}$, the observed dataset $\mathcal{D} = \{(x_n, y_n)\}_{n=1}^{N}$ and any unseen data point

$(\boldsymbol{x}, \boldsymbol{y})$ is factorized as follows:

$$\mathbf{Pr}[\boldsymbol{x}, \boldsymbol{y}, \mathcal{D}, \boldsymbol{\theta}] = (\mathbf{Pr}[\boldsymbol{y}|\boldsymbol{x}, \boldsymbol{\theta}]\mathbf{Pr}[\boldsymbol{x}] \left(\mathbf{Pr}[\mathcal{D}|\boldsymbol{\theta}]\right)) \mathbf{Pr}[\boldsymbol{\theta}] \tag{14}$$

$$= \left(\mathbf{Pr}[\boldsymbol{y}|\boldsymbol{x}, \boldsymbol{\theta}]\mathbf{Pr}[\boldsymbol{x}] \left(\prod_{n=1}^{N} \mathbf{Pr}[y_n|x_n, \boldsymbol{\theta}]\mathbf{Pr}[x_n]\right)\right) \mathbf{Pr}[\boldsymbol{\theta}] \tag{15}$$

where $\mathbf{Pr}[x_i]$'s and $\mathbf{Pr}[\boldsymbol{x}]$ are identically distributed, and so are the conditional distributions $\mathbf{Pr}[y_i|x_i, \boldsymbol{\theta}]$'s and $\mathbf{Pr}[\boldsymbol{y}|\boldsymbol{x}, \boldsymbol{\theta}]$. In other words, we assume that (1) the samples $(x_n, y_n)$'s (and unseen data point $(\boldsymbol{x}, \boldsymbol{y})$) are are identical and independent distributed according to the same data distribution; and (2) $x_n$ (or $\boldsymbol{x}$) and $\boldsymbol{\theta}$ together predict the output $y_n$ (or $\boldsymbol{y}$) according to the same conditional distribution. Notice that the factorization above also implies the following equations:

$$\mathbf{Pr}[\boldsymbol{y}|\boldsymbol{x}, \mathcal{D}, \boldsymbol{\theta}] = \mathbf{Pr}[\boldsymbol{y}|\boldsymbol{x}, \boldsymbol{\theta}] \tag{16a}$$

$$\mathbf{Pr}[\boldsymbol{\theta}|\boldsymbol{x}, \mathcal{D}] = \mathbf{Pr}[\boldsymbol{\theta}|\mathcal{D}] \tag{16b}$$

With these implications, the posterior predictive distribution $\mathbf{Pr}[\boldsymbol{y}|\boldsymbol{x}, \mathcal{D}]$ can now expanded as:

$$\mathbf{Pr}[\boldsymbol{y}|\boldsymbol{x}, \mathcal{D}] = \int_{\theta} \mathbf{Pr}[\boldsymbol{y}|x, \theta, \mathcal{D}]\mathbf{Pr}[\theta|x, \mathcal{D}]d\theta = \int_{\theta} \mathbf{Pr}[\boldsymbol{y}|x, \theta] \underbrace{\mathbf{Pr}[\theta|\mathcal{D}]}_{\approx Q(\theta; \phi)} d\theta \tag{17}$$

where we approximate the posterior distribution $\mathbf{Pr}[\boldsymbol{\theta}|\mathcal{D}]$ by a parameterized distribution $Q(\boldsymbol{\theta}; \phi)$.

**Variational Learning**   The reason we are learning an approximate posterior $Q$ and not the exact distribution $\mathbf{Pr}[\boldsymbol{\theta}|\mathcal{D}]$ is that for complex models the latter is intractable to compute. The exact posterior $\mathbf{Pr}[\boldsymbol{\theta}|\mathcal{D}]$ generally does not take the form of $Q(\boldsymbol{\theta}; \phi)$ even if its prior $\mathbf{Pr}[\boldsymbol{\theta}]$ does.

A standard approach to finding a good approximation $Q(\boldsymbol{\theta}; \phi)$ is *variational inference*, which finds $\phi^{\star}$ such that the *KL-divergence* $\mathbf{KL}(Q(\boldsymbol{\theta}; \phi)||\mathbf{Pr}[\boldsymbol{\theta}|\mathcal{D}])$ of $Q(\boldsymbol{\theta}; \phi)$ from $\mathbf{Pr}[\boldsymbol{\theta}|\mathcal{D}]$ is minimized (or alternatively the negative KL-divergence is maximized.)

$$\phi^{\star} = \arg\max_{\phi} \left(-\mathbf{KL}(Q(\boldsymbol{\theta}; \phi)||\mathbf{Pr}[\boldsymbol{\theta}|\mathcal{D}])\right) \tag{18}$$

$$= \arg\max_{\phi} \left(-\int_{\theta} Q(\theta; \phi) \log\left(\frac{Q(\theta; \phi)}{\mathbf{Pr}[\theta|\mathcal{D}]}\right) d\theta\right) \tag{19}$$

where $\mathbf{Pr}[\boldsymbol{\theta}|\mathcal{D}]$ is obtained via standard *Bayes' rule*, i.e. $\mathbf{Pr}[\boldsymbol{\theta}|\mathcal{D}] = \mathbf{Pr}[\mathcal{D}|\boldsymbol{\theta}]\mathbf{Pr}[\boldsymbol{\theta}]/\mathbf{Pr}[\mathcal{D}]$. Now we are able to decompose the maximization objective into two terms by plugging the rule into (19):

$$\mathcal{L}(\phi) = -\int_{\theta} Q(\theta; \phi) \log\left(Q(\theta; \phi) \cdot \frac{\mathbf{Pr}[\mathcal{D}]}{\mathbf{Pr}[\theta]\mathbf{Pr}[\mathcal{D}|\theta]}\right) d\boldsymbol{\theta} \tag{20}$$

$$= \sum_{n=1}^{N} \int_{\theta} \log\left(\mathbf{Pr}[y_n|x_n, \theta]\right) Q(\theta; \phi)d\theta + \int_{\theta} Q(\theta; \phi) \log\left(\frac{Q(\theta; \phi)}{\mathbf{Pr}[\theta]}\right) d\theta + \text{const.} \tag{21}$$

$$= \sum_{n=1}^{N} \underbrace{\mathbb{E}_Q\left[\log\left(\mathbf{Pr}[y_n|x_n, \theta]\right)\right]}_{\mathcal{L}_n(\phi)} + \underbrace{\mathbf{KL}(Q(\boldsymbol{\theta}; \phi)||\mathbf{Pr}[\boldsymbol{\theta}])}_{\mathcal{R}(\phi)} - \underbrace{\log\left(\mathbf{Pr}[\mathcal{D}]\right)}_{\text{const.}} \tag{22}$$

where (1) $\mathcal{L}_n(\phi)$ is the *expected log-likelihood*, which reflects the predictive performance of the Bayesian model on the data point $(x_n, y_n)$; and (2) $\mathcal{R}(\phi)$ is the KL-divergence between $Q(\boldsymbol{\theta}; \phi)$ and its prior $\mathbf{Pr}[\boldsymbol{\theta}]$, which reduces to entropy $H(Q)$ if the prior of $\boldsymbol{\theta}$ follows a uniform distribution.

**Hierarchical Bayesian Model**   A Bayesian neural network can be considered as a hierarchical Bayesian model depicted in Figure 3b, which further satisfies the following two assumptions:

**Assumption B.1 (Independence of Model Parameters $\boldsymbol{\theta}^{(l)}$).** *The approximate posterior $Q(\boldsymbol{\theta}; \phi)$ over the model parameters $\boldsymbol{\theta}$ are partitioned into $L$ disjoint and statistically independent layers $\{\boldsymbol{\theta}^{(l)}\}_{l=0}^{L-1}$ (where each $\phi^{(l)}$ parameterizes $\boldsymbol{\theta}^{(l)}$ in the l-th layer) such that:*

$$Q(\boldsymbol{\theta}; \phi) = \prod_{l=0}^{L-1} Q(\boldsymbol{\theta}^{(l)}; \phi^{(l)}) \tag{23}$$

**Assumption B.2 (Markovianity of Hidden Units $h^{(l)}$).** *The hidden variables $h = \{h^{(l)}\}_{l=0}^L$ satisfy the* Markov *property that $h^{(l+1)}$ depends on the input $x$ only through its previous layer $h^{(l)}$:*

$$\mathbf{Pr}[h^{(l+1)}|h^{(:l)}, \theta^{(:l)}] = \mathbf{Pr}[h^{(l+1)}|h^{(l)}, \theta^{(l)}] \tag{24}$$

*where we use short-hand notations $h^{(:l)}$ and $\theta^{(:l)}$ to represent the sets of previous layers $\{h^{(k)}\}_{k=0}^l$ and $\{\theta^{(k)}\}_{k=0}^l$. For consistency, we denote $h^{(0)} = x$ and $h^{(L)} = y$.*

**Proof of probabilistic prorogation**    Based on the two assumptions above, we provide a proof for *probabilistic propagation* in Equation (1) as follows:

$$\overbrace{\mathbf{Pr}[h^{(l+1)}|x]}^{P\left(h^{(l+1)}; \psi^{(l+1)}\right)} = \int_{\theta^{(:l)}} \mathbf{Pr}[h^{(l+1)}|x, \theta^{(:l)}] \, Q(\theta^{(:l)}; \phi^{(:l)}) \, d\theta^{(:l)} \tag{25}$$

$$= \int_{\theta^{(:l)}} \left( \int_{h^{(l)}} \mathbf{Pr}[h^{(l+1)}|h^{(l)}, \theta^{(l)}] \mathbf{Pr}[h^{(l)}|x, \theta^{(:l-1)}] dh^{(l)} \right) Q(\theta^{(:l)}; \phi^{(:l)}) \, d\theta^{(:l)} \tag{26}$$

$$= \int_{h^{(l)}, \theta^{(l)}} \mathbf{Pr}[h^{(l+1)}|h^{(l)}, \theta^{(l)}] Q(\theta^{(l)}; \phi^{(l)}) \\ \left( \int_{\theta^{(:l-1)}} \mathbf{Pr}[h^{(l)}|x, \theta^{(:l-1)}] Q(\theta^{(:l-1)}; \phi^{(:l-1)}) d\theta^{(:l-1)} \right) dh^{(l)} d\theta^{(l)} \tag{27}$$

$$= \int_{h^{(l)}, \theta^{(l)}} \mathbf{Pr}[h^{(l+1)}|h^{(l)}, \theta^{(l)}] Q(\theta^{(l)}; \phi^{(l)}) \underbrace{\mathbf{Pr}[h^{(l)}|x]}_{P\left(h^{(l)}; \psi^{(l)}\right)} dh^{(l)} d\theta^{(l)} \tag{28}$$

# C    ALTERNATIVE EVIDENCE LOWER BOUND AND ITS ANALYTIC FORMS

## C.1    ALTERNATIVE EVIDENCE LOWER BOUND (PROOF FOR THEOREM 3.1)

The steps to prove the inequality (6) almost follow the ones for probabilistic propagation above:

$$\mathcal{L}_n(\phi) = \mathbb{E}_Q \left[ \log(\mathbf{Pr}[y_n|x_n, \theta]) \right] \tag{29}$$

$$= \int_\theta \log\left(\mathbf{Pr}[y_n|x_n, \theta]\right) Q(\theta; \phi) d\theta \tag{30}$$

$$= \int_\theta \log\left( \int_{h^{(L-1)}} \mathbf{Pr}[y_n, h^{(L-1)}|x_n, \theta] dh^{(L-1)} \right) Q(\theta; \phi) d\theta \tag{31}$$

$$= \int_\theta \log\left( \int_{h^{(L-1)}} \mathbf{Pr}[y_n|h^{(L-1)}, \theta^{(L-1)}] \mathbf{Pr}[h^{(L-1)}|x_n, \theta^{(0:L-2)}] dh^{(L-1)} \right) Q(\theta; \phi) d\theta \tag{32}$$

$$\geq \int_\theta \left( \int_{h^{(L-1)}} \log\left( \mathbf{Pr}[y_n|h^{(L-1)}, \theta^{(L-1)}] \right) \mathbf{Pr}[h^{(L-1)}|x_n, \theta^{(0:L-1)}] dh^{(L-1)} \right) Q(\theta; \phi) d\theta \tag{33}$$

$$= \int_{h^{(L-1)}, \theta^{(L-1)}} \log\left( \mathbf{Pr}[y_n|h^{(L-1)}, \theta^{(L-1)}] \right) Q(\theta^{(L-1)}; \phi^{(L-1)}) \\ \left( \int_{\theta^{(0:L-2)}} \mathbf{Pr}[h^{(L-1)}|x_n, \theta^{(0:L-2)}] Q(\theta^{(0:L-2)}; \phi^{(0:L-2)}) d\theta^{(0:L-2)} \right) dh^{(L-1)} d\theta^{(L-1)} \tag{34}$$

$$= \int_{h^{(L-1)}, \theta^{(L-1)}} \log\left( \mathbf{Pr}[y_n|h^{(L-1)}, \theta^{(L-1)}] \right) Q(\theta^{(L-1)}) \mathbf{Pr}[h^{(L-1)}|x_n] dh^{(L-1)} d\theta^{(L-1)} \tag{35}$$

$$= \mathbb{E}_{h^{(L-1)} \sim P; \, \theta^{(L-1)} \sim Q} \left[ \log\left( \mathbf{Pr}[y_n|h^{(L-1)}, \theta^{(L-1)}] \right) \right] = \overline{\mathcal{L}}_n(\phi) \tag{36}$$

where the key is the Jensen's inequality $\mathbb{E}_Q \left[ \log(\cdot) \right] \geq \log\left( \mathbb{E}_Q \left[ \cdot \right] \right)$ in Equation (33). Notice that if $\theta^{(L-1)}$ is not random variable (typical for an output layer), $\overline{\mathcal{L}}_n(\phi)$ can be simplified as:

$$\overline{\mathcal{L}}_n(\phi) = \int_{h^{(L-1)}} \log\left( \mathbf{Pr}[y_n|h^{(L-1)}; \phi^{(L-1)}] \right) P(h^{(L-1)}; \psi^{(L-1)}) dh^{(L-1)} \tag{37}$$

where we write $\mathbf{Pr}[h^{(L-1)}|x]$ in its parameterized form $P(h^{(L-1)}; \psi^{(L-1)})$. Now, the gradient $\partial \overline{\mathcal{L}}_n(\phi)/\partial \phi^{(L-1)}$ can be obtained by differentiating over Equation (37), while other gradients $\partial \overline{\mathcal{L}}_n(\phi)/\phi^{(:L-2)}$ further obtained by chain rule:

$$\frac{\partial \overline{\mathcal{L}}_n(\phi)}{\partial \phi^{(:L-2)}} = \frac{\partial \overline{\mathcal{L}}_n(\phi)}{\partial \psi^{(L-1)}} \cdot \frac{\partial \psi^{(L-1)}}{\partial \phi^{(:L-2)}} \tag{38}$$

which requires us to compute $\partial \overline{\mathcal{L}}_n(\phi)/\partial \psi^{(L-1)}$ and $\partial \psi^{(L-1)}/\partial \phi^{(:L-2)}$. While $\partial \overline{\mathcal{L}}_n(\phi)/\partial \psi^{(L-1)}$ can be derived from Equation (37), $\partial \psi^{(L-1)}/\partial \phi^{(:L-2)}$ can be obtained by backpropagating outputs of the $(L-2)^{\text{th}}$ layer obtained from probabilistic propagation in Equation (1). In other words: since $P(h^{(L-1)}; \psi^{(L-1)})$ is an intermediate step of the forward pass, $\psi^{(L-1)}$ is a function of all parameters from previous layers $\phi^{(:L-2)}$, and if each step $\psi^{(l+1)} = g^{(l)}(\psi^{(l)}, \phi^{(l)})$ is differentiable w.r.t. $\psi^{(l)}$ and $\phi^{(l)}$, the partial derivatives $\partial \psi^{(L-1)}/\partial \phi^{(:L-2)}$ can be obtained by iterative chain rule.

## C.2 Softmax Layer for Classification Problem

In this part, we first prove the alternative evidence lower bound (ELBO) for Bayesian neural networks with *softmax* function as their last layers. Subsequently, we derive the corresponding backpropagation rule for the softmax layer. Finally, we show a method based on *Taylor's expansion* to approximately evaluate a softmax layer without *Monte Carlo sampling*.

**Theorem C.1 (Analytic Form of $\overline{\mathcal{L}}_n(\phi)$ for Classification).** *Let $h \in \mathbb{R}^K$ (with K the number of classes) be the pre-activations of a softmax layer (a.k.a. logits), and $\phi = s \in \mathbb{R}^+$ be a scaling factor that adjusts its scale such that $\mathbf{Pr}[y = c|h, s] = \exp(h_c/s)/\sum_{k=1}^K \exp(h_k/s)$. Suppose the logits $\{h_k\}_{k=1}^K$ are pairwise independent (which holds under mean-field approximation) and $h_k$ follows a Gaussian distribution $h_k \sim \mathcal{N}(\mu_k, \nu_k)$ (therefore $\psi = \{\mu_k, \nu_k\}_{k=1}^K$) and s is a deterministic parameter. Then $\overline{\mathcal{L}}_n(\phi)$ can be further upper bound by the following analytic form:*

$$\overline{\mathcal{L}}_n(\phi) \geq \frac{\mu_c}{s} - \log \left( \sum_{k=1}^K \exp \left( \frac{\mu_k}{s} + \frac{\nu_k}{2s^2} \right) \right) \triangleq \hat{\mathcal{L}}(\phi) \tag{39}$$

*Proof.* The lower bound follows by plugging $\mathbf{Pr}[y|h, s]$ and $\mathbf{Pr}[h_k|x]$ into Equation (6).

$$\overline{\mathcal{L}}_n(\phi) = \int_h \log \left( \mathbf{Pr}[y_n = c|h; s] \right) \mathbf{Pr}[h|x] dh \tag{40}$$

$$= \int_h \left( \frac{h_c}{s} - \log \left( \sum_{k=1}^K \exp \left( \frac{h_k}{s} \right) \right) \right) \left( \prod_{k=1}^K \mathbf{Pr}[h_k|x] \right) dh \tag{41}$$

$$= \frac{1}{s} \int_{h_c} h_c \mathbf{Pr}[h_c|x_n] dh_c - \int_h \log \left( \sum_{k=1}^K \exp \left( \frac{h_k}{s} \right) \right) \left( \prod_{k=1}^K \mathbf{Pr}[h_k|x] \right) dh \tag{42}$$

$$= \frac{\mu_c}{s} - \int_h \log \left( \sum_{k=1}^K \exp \left( \frac{h_k}{s} \right) \right) \left( \prod_{k=1}^K \mathbf{Pr}[h_k|x] \right) dh \tag{43}$$

$$\geq \frac{\mu_c}{s} - \log \left( \int_h \sum_{k=1}^K \exp \left( \frac{h_k}{s} \right) \left( \prod_{k=1}^K \mathbf{Pr}[h_k|x] \right) dh \right) \tag{44}$$

$$= \frac{\mu_c}{s} - \log \left( \sum_{k=1}^K \int_{h_k} \exp \left( \frac{h_k}{s} \right) \mathbf{Pr}[h_k|x] dh_k \right) \tag{45}$$

$$= \frac{\mu_c}{s} - \log \left( \sum_{k=1}^K \int_{h_k} \exp \left( \frac{h_k}{s} \right) \cdot \frac{1}{\sqrt{2\pi\nu_k}} \exp \left( -\frac{(h_k - \mu_k)^2}{2\nu_k} \right) dh_k \right) \tag{46}$$

$$= \frac{\mu_c}{s} - \log \left( \sum_{k=1}^K \exp \left( \frac{\mu_k}{s} + \frac{\nu_k}{2s^2} \right) \right) = \hat{\mathcal{L}}(\phi) \tag{47}$$

where the last equation follows

$$\int_{h_k} \exp\left(\frac{h_k}{s}\right) \cdot \frac{1}{\sqrt{2\pi\nu_k}} \exp\left(-\frac{(h_k - \mu_k)^2}{2\nu_k}\right) dh_k \tag{48}$$

$$= \int_{h_k} \frac{1}{\sqrt{2\pi\nu_k}} \exp\left(-\frac{h_k^2 - 2(\mu_k + \nu_k/s)h_k + \mu_k^2}{2\nu_k}\right) dh_k \tag{49}$$

$$= \underbrace{\int_{h_k} \frac{1}{\sqrt{2\pi\nu_k}} \exp\left(-\frac{(h_k - (\mu_k + \nu_k))^2}{2\nu_k}\right) dh_k} \cdot \exp\left(\frac{\mu_k}{s} + \frac{\nu_k}{2s^2}\right) \tag{50}$$

where the under-braced term is unity since it takes the form of Gaussian distribution. $\qquad\square$

From Equation (43) to (44), we use the Jensen's inequality to achieve a lower bound for integral of log-sum-exp. The bound can be tighten with advanced techniques in Khan (2012).

**Derivatives of $\overline{\mathcal{L}}_n(\phi)$ in** (39)    To use probabilistic backpropagation to obtain the gradients w.r.t. the parameters from previous layers, we first need to obtain the derivatives w.r.t. $\psi^{(L-1)} = \{\mu_k, \nu_k\}_{k=1}^K$.

$$\frac{\partial \hat{\mathcal{L}}_n(\phi)}{\partial \mu_k} = -\frac{1}{s}\left(\frac{\exp\left(\mu_k/s + \nu_k/2s^2\right)}{\sum_{k=1}^K \exp\left(\mu_k/s + \nu_k/2s^2\right)} - \mathbf{1}[k = c]\right) \tag{51a}$$

$$\frac{\partial \hat{\mathcal{L}}_n(\phi)}{\partial \nu_k} = -\frac{1}{2s^2}\left(\frac{\exp\left(\mu_k/s + \nu_k/2s^2\right)}{\sum_{k=1}^K \exp\left(\mu_k/s + \nu_k/2s^2\right)}\right) \tag{51b}$$

Furthermore, the scale $s$ can be (optionally) updated along with other parameters using the gradient

$$\frac{\partial \hat{\mathcal{L}}_n(\phi)}{\partial s} = -\frac{\mu_c}{s^2} + \frac{\sum_{k=1}^K \left(\mu_k/s^2 + \nu_k/s^3\right) \exp\left(\mu_k/s + \nu_k/2s^2\right)}{\sum_{k=1}^K \exp\left(\mu_k/s + \nu_k/2s^2\right)} \tag{52}$$

**Prediction with Softmax Layer**    Once we learn the parameters for the Bayesian neural network, in principle we can compute the predictive distribution of $\boldsymbol{y}$ by evaluating the following equation:

$$\mathbf{Pr}[\boldsymbol{y} = c|\boldsymbol{x}] = \int_h \mathbf{Pr}[\boldsymbol{y} = c|\boldsymbol{h}, s]\mathbf{Pr}[h|\boldsymbol{x}]dh = \int_h \ell_c(h)\mathbf{Pr}[h|\boldsymbol{x}]dh \tag{53}$$

$$\text{(Mean-field assumption)} = \int_{h_1}\cdots\int_{h_K} \ell_c(h)\left(\prod_{k=1}^K \mathbf{Pr}[h_k|x]\right) dh_1\cdots dh_k \tag{54}$$

where we denote the softmax function as $\ell_c(h) = \exp(h_c/s)/[\sum_k \exp(h_k/s)]$. Unfortunately, the equation above can not be computed in closed form. The most straight-forward work-around is to approximate the integral by Monte Carlo sampling: for each $\boldsymbol{h}_k$ we draw $S$ samples $\{h_k^s\}_{s=1}^S$ independently and compute the prediction:

$$\mathbf{Pr}[\boldsymbol{y} = c|x] \approx \frac{1}{S}\sum_{s=1}^S \ell_c(h^s), \ \forall c \in [K] \tag{55}$$

Despite its conceptual simplicity, Monte Carlo method suffers from expensive computation and high variance in estimation. Instead, we propose an economical estimate based on Taylor's expansion. First, we expand the function $\ell_c(h)$ by Taylor's series at the point $\mu$ (up to the second order):

$$\ell_c(h) = \ell_c(\mu) + \left[\frac{\partial \ell_c}{\partial h}(\mu)\right]^\top (h - \mu) + \frac{1}{2}(h - \mu)^\top \left[\frac{\partial^2 \ell_c}{\partial h^2}(\mu)\right](h - \mu) + O\left(\|h - c\|^3\right) \tag{56}$$

$$= \ell_c(\mu) + \sum_{k=1}^K \left[\frac{\partial \ell_c}{\partial h_k}(\mu)\right](h_k - \mu_k) + \sum_{i=1}^K \sum_{j=1}^K \left[\frac{\partial^2 \ell_c}{\partial h_i h_j}(\mu)\right](h_i - \mu_i)(h_j - \mu_j) + O\left(\|h - \mu\|^3\right)$$

$$\tag{57}$$

Before we derive the forms of these derivatives, we first show the terms of odd orders do not contribute to the expectation. For example, if $\ell_c(h)$ is approximated by its first two terms (i.e. a linear function), Equation (54) can be written as

$$\mathbf{Pr}[\boldsymbol{y} = c | x] \approx \int_{h_1} \cdots \int_{h_K} \left( \ell_c(\mu) + \sum_{k=1}^K \left[ \frac{\partial \ell_c}{\partial h_k}(\mu) \right] (h_k - \mu_k) \right) \left( \prod_{k=1}^K \mathbf{Pr}[h_k | x] \right) dh_1 \cdots dh_k \tag{58}$$

$$= \ell_c(\mu) + \sum_{k=1}^K \left[ \frac{\partial \ell_c}{\partial h_k}(\mu) \right] \left( \int_{h_k} (h_k - \mu_k) \mathbf{Pr}[h_k | x] dh_k \right) = \ell_c(\mu) \tag{59}$$

where the second term is zero by the symmetry of $\mathbf{Pr}[\boldsymbol{h}_k | \boldsymbol{x}]$ around $\mu_k$ (or simply the definition of $\mu_k$'s). Therefore, the first-order approximation results exactly in a (deterministic) softmax function of the mean vector $\mu$. In order to incorporate the variance into the approximation, we will need to derive the exact forms of the derivatives of $\ell_c(h)$. Specifically, the first-order derivatives are obtained from the definition of $\ell_c(h)$.

$$\frac{\partial \ell_c}{\partial h_c}(h) = \frac{1}{s} \cdot \frac{\exp(h_c/s) - \exp(2h_c/s)}{\left( \sum_{k=1}^K \exp(h_k/s) \right)^2} = \frac{1}{s} \left( \ell_c(h) - \ell_c^2(h) \right) \tag{60a}$$

$$\frac{\partial \ell_c}{\partial h_k}(h) = -\frac{1}{s} \cdot \frac{\exp(h_c/s) \cdot \exp(h_k/s)}{\left( \sum_{k=1}^K \exp(h_k/s) \right)^2} = -\frac{1}{s} \ell_c(h) \ell_k(h), \ \forall k \neq c \tag{60b}$$

and subsequently the second-order derivatives from the first ones:

$$\frac{\partial^2 \ell_c}{\partial h_c^2}(h) = \frac{1}{s} \left( \frac{\partial \ell_c}{\partial h_c}(h) - 2\ell_c(h) \frac{\partial \ell_c}{\partial h_c}(h) \right) = \frac{1}{s^2} \left( \ell_c(h) - 3\ell_c^2(h) + 2\ell_c^3(h) \right) \tag{61a}$$

$$\frac{\partial^2 \ell_c}{\partial h_k^2}(h) = -\frac{1}{s} \left( \frac{\partial \ell_c}{\partial h_c}(h) \ell_k(h) + \ell_c(h) \frac{\partial \ell_k}{\partial h_c}(h) \right) = \frac{1}{s^2} \left( -\ell_c(h) \ell_k(h) + 2\ell_c^2(h) \ell_k(h) \right), \ \forall k \neq c \tag{61b}$$

with these derivatives we can compute the second-order approximation as

$$\mathbf{Pr}[\boldsymbol{y} = c | x] \approx \int_{h_1, \cdot, h_K} \left( \ell_c(\mu) + \frac{1}{2} \sum_{i=1}^K \sum_{j=1}^K \frac{\partial^2 \ell_c}{\partial \mu_i \mu_j}(\mu)(h_i - \mu_i)(h_j - \mu_j) \right) \left( \prod_{k=1}^K \mathbf{Pr}[h_k | x] \right) dh_1 \cdots dh_K \tag{62}$$

$$= \ell_c(\mu) + \frac{1}{2} \frac{\partial^2 \ell_c}{\partial \mu_c^2}(\mu) \int_{h_c} (h_c - \mu_c)^2 \mathbf{Pr}[h_c | x] dh_c + \frac{1}{2} \sum_{k \neq c} \frac{\partial^2 \ell_c}{\partial \mu_k^2}(\mu) \int_{h_k} (h_k - \mu_k)^2 \mathbf{Pr}[h_k | x] dh_k \tag{63}$$

$$= \ell_c(\mu) + \frac{1}{2s^2} \left( \ell_c(\mu) - 3\ell_c^2(\mu) + 2\ell_c^3(\mu) \right) \nu_c + \frac{1}{2s^2} \sum_{k \neq c} \left( -\ell_c(\mu) \ell_k(\mu) + 2\ell_c^2(\mu) \ell_k(\mu) \right) \nu_k \tag{64}$$

$$= \ell_c(\mu) + \frac{1}{2s^2} \left( \ell_c(\mu) - 2\ell_c^2(\mu) \right) \left( \nu_c - \sum_{k=1}^K \ell_k(\mu) \nu_k \right) \tag{65}$$

The equation above can be further written in vector form as:

$$\mathbf{Pr}[\boldsymbol{y} | x] \approx \ell(\mu) + \frac{1}{2s^2} \left( \ell(\mu) - \ell(\mu)^{\circ 2} \right) \circ \left( \nu - \ell(\mu)^\top \nu \right) \tag{66}$$

## C.3  GAUSSIAN OUTPUT LAYER FOR REGRESSION PROBLEM

In this part, we develop an alternative evidence lower bound (ELBO) for Bayesian neural networks with Gaussian output layers, and derive the corresponding gradients for backpropagation. Despite the difficulty to obtain an analytical predictive distribution for the output, we show that its *central moments* can be easily computed given the learned parameters.

**Theorem C.2 (Analytic Form of $\overline{\mathcal{L}}_n(\phi)$ for Regression).** *Let $h \in \mathbb{R}^I$ be the output of last hidden layer (with $I$ the number of hidden units), and $\phi = (w, s) \in \mathbb{R}^I \times \mathbb{R}^+$ be the parameters that define the predictive distribution over output $y$ as*

$$\mathbf{Pr}[y|h; w, s] = \frac{1}{\sqrt{2\pi s}} \exp\left(-\frac{(y - w^\top h)^2}{2s}\right) \tag{67}$$

*Suppose the hidden units $\{h_k\}_{k=1}^K$ are pairwise independent (which holds under mean-field approximation), and each $h_i$ has mean $\mu_i$ and variance $\nu_i$, then $\overline{\mathcal{L}}_n(\phi)$ takes an analytic form:*

$$\overline{\mathcal{L}}_n(\phi) = -\frac{(y - w^\top \mu)^2 + (w^{\circ 2})^\top \nu}{2s} - \frac{\log(2\pi s)}{2} \tag{68}$$

*where $(w^{\circ 2})_i = w_i^2$ and $\mu = [\mu_1, \cdots, \mu_I]^\top \in \mathbb{R}^I$ and $\nu = [\nu_1, \cdots, \nu_I]^\top \in \mathbb{R}^I$ are vectors of mean and variance of the hidden units $h$.*

*Proof.* The Equation (68) is obtained by plugging $\mathbf{Pr}[y|h; w, s]$ into Equation (6).

$$\overline{\mathcal{L}}_n(\phi) = \sum_{h_1} \cdots \sum_{h_I} \log\left(\mathbf{Pr}[y|h_1, \cdots, h_I; w, s]\right)\left(\prod_{i=1}^I \mathbf{Pr}[h_i|x_n]\right) \tag{69}$$

$$= -\sum_{h_1} \cdots \sum_{h_I} \left(\frac{\left(y - \sum_{i=1}^I w_i h_i\right)^2}{2s} + \frac{\log(2\pi s)}{2}\right)\left(\prod_{i=1}^I \mathbf{Pr}[h_i|x_n]\right) \tag{70}$$

$$= -\frac{1}{2s}\sum_{h_1} \cdots \sum_{h_I} \left(y - \sum_{i=1}^I w_i h_i\right)^2 \left(\prod_{i=1}^I \mathbf{Pr}[h_i|x_n]\right) - \frac{\log(2\pi s)}{2} \tag{71}$$

where the long summation in the first term can be further simplified with notations of $\mu$ and $\nu$:

$$\sum_{h_1} \cdots \sum_{h_I} \left(y - \sum_{i=1}^I w_i h_i\right)^2 \left(\prod_{i=1}^I \mathbf{Pr}[h_i|x_n]\right) \tag{72}$$

$$= \sum_{h_1} \cdots \sum_{h_I} \left(y^2 - 2y\sum_{i=1}^I w_i h_i + \sum_{i=1}^I w_i^2 h_i^2 + \sum_{j=1}^I \sum_{k \neq j} w_j w_k h_j h_k\right)\left(\prod_{i=1}^I \mathbf{Pr}[h_i|x_n]\right) \tag{73}$$

$$= y^2 - 2y\sum_{i=1}^I w_i \left(\sum_{h_i} h_i \mathbf{Pr}[h_i|x]\right) + \sum_{i=1}^I w_i^2 \left(\sum_{h_i} h_i^2 \mathbf{Pr}[h_i|x_n]\right)$$
$$+ \sum_{j=1}^I \sum_{k \neq j} w_j w_k \left(\sum_{h_j} h_j \mathbf{Pr}[h_j|x_n]\right)\left(\sum_{h_k} h_k \mathbf{Pr}[h_k|x_n]\right) \tag{74}$$

$$= y^2 - 2y\sum_{i=1}^I w_i \mu_i + \sum_{i=1}^I w_i^2(\mu_i^2 + \nu_i) + \sum_{j=1}^I \sum_{k \neq j} w_j w_k \mu_j \mu_k \tag{75}$$

$$= y^2 - 2y\sum_{i=1}^I w_i \mu_i + \sum_{i=1}^I w_i^2 \nu_i + \left(\sum_{j=1}^I w_j \mu_j\right)\left(\sum_{k=1}^I w_k \mu_k\right) \tag{76}$$

$$= y^2 - 2y\, w^\top \mu + (w^{\circ 2})^\top \nu + \left(w^\top \mu\right)^2 \tag{77}$$

$$= (y - w^\top \mu)^2 + (w^{\circ 2})^\top \nu \tag{78}$$

where $w^{\circ 2}$ denotes element-wise square, i.e. $w^{\circ 2} = [w_1^2, \cdots, w_I^2]^\top$. $\qquad\square$

**Derivatives of $\overline{\mathcal{L}}_n(\phi)$ in Equation** (68)    It is not difficult to show that the gradient of $\overline{\mathcal{L}}_n(\phi)$ can be backpropagated through the last layer. by computing derivatives of $\overline{\mathcal{L}}_n(\phi)$ w.r.t. $\mu$ and $\nu$:

$$\frac{\partial \overline{\mathcal{L}}_n(\phi)}{\partial \mu} = -\frac{(y - w^\top \mu)w}{s} \tag{79a}$$

$$\frac{\partial \overline{\mathcal{L}}_n(\phi)}{\partial \nu} = -\frac{w^{\circ 2}}{2s} \tag{79b}$$

Furthermore, the parameters $\{w, s\}$ can be updated along with other parameters with their gradients:

$$\frac{\partial \overline{\mathcal{L}}_n(\phi)}{\partial w} = -\frac{(y - w^\top \mu)\mu + (w \circ \nu)}{s} \tag{80a}$$

$$\frac{\partial \overline{\mathcal{L}}_n(\phi)}{\partial s} = -\frac{1}{2s} + \frac{(y - w^\top \mu)^2 + (w^{\circ 2})^\top \nu}{2s^2} \tag{80b}$$

**Prediction with Gaussian Layer**    Once we determine the parameters for the last layer, in principle we can compute the predictive distribution $\mathbf{Pr}[\boldsymbol{y}|x]$ for the output $\boldsymbol{y}$ given the input $x$ according to

$$\mathbf{Pr}[\boldsymbol{y}|x] = \sum_h \mathbf{Pr}[\boldsymbol{y}|h; w, s]\mathbf{Pr}[h|\boldsymbol{x}] = \sum_{h_1} \cdots \sum_{h_I} \mathbf{Pr}[\boldsymbol{y}|h; w, s] \left( \prod_{i=1}^I \mathbf{Pr}[h_i|x] \right)$$

$$= \sum_{h_1} \cdots \sum_{h_I} \frac{1}{\sqrt{2\pi s}} \exp \left( -\frac{\left( \boldsymbol{y} - \sum_{i=1}^I w_i h_i \right)^2}{2s} \right) \left( \prod_{i=1}^I \mathbf{Pr}[h_i|x] \right) \tag{81}$$

Unfortunately, exact computation of the equation above for arbitrary output value $y$ is intractable in general. However, the central moments of the predictive distribution $\mathbf{Pr}[\boldsymbol{y}|x]$ are easily evaluated. Consider we interpret the prediction as $\boldsymbol{y} = w^\top \boldsymbol{h} + \boldsymbol{\epsilon}$, where $\boldsymbol{\epsilon} \sim \mathcal{N}(0, s)$, its mean and variance can be easily computed as

$$\mathbb{E}[\boldsymbol{y}|x] = w^\top \mathbb{E}[\boldsymbol{h}] = w^\top \mu \tag{82a}$$

$$\mathbb{V}[\boldsymbol{y}|x] = (w^{\circ 2})^\top \mathbb{V}[\boldsymbol{h}] + \mathbb{V}[\boldsymbol{\epsilon}] = (w^{\circ 2})^\top \nu + s \tag{82b}$$

Furthermore, if we denote the (normalized) *skewness* and *kurtosis* of $\boldsymbol{h}_i$ as $\gamma_i$ and $\kappa_i$:

$$\gamma_i = \mathbb{E}\left[ (\boldsymbol{h}_i - \mu_i)^3 | x \right] / \nu_i^{3/2} = \sum_{h_i} (h_i - \mu_i)^3 \mathbf{Pr}[h_i|x] / \nu_i^{3/2} \tag{83a}$$

$$\kappa_i = \mathbb{E}\left[ (\boldsymbol{h}_i - \mu_i)^4 | x \right] / \nu_i^2 \;\;= \sum_{h_i} (h_i - \mu_i)^4 \mathbf{Pr}[h_i|x] / \nu_i^2 \tag{83b}$$

Then the (normalized) skewness and kurtosis of the prediction $\boldsymbol{y}$ are also easily computed with the vectors of $\gamma = [\gamma_1, \cdots, \gamma_I]^\top \in \mathbb{R}^I$ and $\kappa = [\kappa_1, \cdots, \kappa_I] \in \mathbb{R}^I$.

$$\gamma[\boldsymbol{y}|x] = \frac{\mathbb{E}\left[ (\boldsymbol{y} - w^\top \mu)^3 | x \right]}{\mathbb{V}[\boldsymbol{y}|x]^{3/2}} = \frac{(w^{\circ 3})^\top (\gamma \circ \nu^{\circ 3/2})}{[(w^{\circ 2})^\top \nu + s]^{3/2}} \tag{84a}$$

$$\kappa[\boldsymbol{y}|x] = \frac{\mathbb{E}\left[ (\boldsymbol{y} - w^\top \mu)^4 | x \right]}{\mathbb{V}[\boldsymbol{y}|x]^2} = \frac{(w^{\circ 4})^\top (\kappa \circ \nu^{\circ 2}) + s(w^{\circ 2})^\top \nu}{[(w^{\circ 2})^\top \nu + s]^2} \tag{84b}$$

# D   PROBABILISTIC PROPAGATION IN BAYESIAN QUANTIZED NETWORKS

In this section, we present fast(er) algorithms for sampling-free probabilistic propagation (i.e. evaluating Equation (8)). According to Section 4, we divide this section into three parts, each part for a specific range of fan-in numbers $E$.

## D.1   SMALL FAN-IN LAYERS: DIRECT TENSOR CONTRACTION

If $E$ is small, tensor contraction in Equation (8) is immediately applicable. Representative layers of small $E$ are *shortcut layer (a.k.a. skip-connection)* and what we name as *depth-wise layers*.

**Shortcut Layer**  With a skip connection, the output $\boldsymbol{h}^{(l+1)}$ is an addition of two previous layers $\boldsymbol{h}^{(l)}$ and $\boldsymbol{h}^{(m)}$. Therefore and the distribution of $\boldsymbol{h}^{(l+1)}$ can be directly computed as

$$P(\boldsymbol{h}_i^{(l+1)}; \psi_i^{(l+1)}) = \sum_{h_i^{(l)}, h_i^{(m)}} \delta[\boldsymbol{h}_i^{(l+1)} = h_i^{(l)} + h_i^{(m)}] \, P(h_i^{(l)}; \psi_i^{(l)}) \, P(h_i^{(m)}; \psi_i^{(m)}) \tag{85}$$

**Depth-wise Layers**  In a depth-wise layer, each output unit $\boldsymbol{h}_i^{(l+1)}$ is a transformation (parameterized by $\boldsymbol{\theta}_i^{(l)}$) of its corresponding input $\boldsymbol{h}_i^{(l)}$, i.e.

$$P(\boldsymbol{h}_i^{(l+1)}; \psi_i^{(l+1)}) = \sum_{h_i^{(l)}, \theta_i^{(l)}} \mathbf{Pr}[\boldsymbol{h}_i^{(l+1)} | h_i^{(l)}, \theta_i^{(l)}] \, Q(\theta_i^{(l)}; \phi_i^{(m)}) \, P(h_i^{(l)}; \psi_i^{(l)}) \tag{86}$$

Depth-wise layers include *dropout layers* (where $\boldsymbol{\theta}^{(l)}$ are dropout rates), *nonlinear layers* (where $\boldsymbol{\theta}^{(l)}$ are threshold values) or *element-wise product layers* (where $\boldsymbol{\theta}^{(l)}$ are the weights). For both shortcut and depth-wise layers, the time complexity is $O(JD^2)$ since $E <= 2$.

### D.2  MEDIUM FAN-IN LAYERS: DISCRETE FOURIER TRANSFORM

In neural networks, representative layers with medium fan-in number $E$ are pooling layers, where each output unit depends on a medium number of input units. Typically, the special structure of pooling layers allows for faster algorithm than computing Equation (8) directly.

**Max and Probabilistic Pooling**  For each output, **(1)** a max pooling layer picks the maximum value from corresponding inputs, i.e. $\boldsymbol{h}_j^{(l+1)} = \max_{i \in \mathcal{I}(j)} \boldsymbol{h}_i^{(l)}$, while **(2)** a probabilistic pooling layer selects the value the inputs following a categorical distribution, i.e. $\mathbf{Pr}[\boldsymbol{h}_j^{(l+1)} = \boldsymbol{h}_i^{(l)}] = \theta_i$. For both cases, the predictive distribution of $\boldsymbol{h}_j^{(l+1)}$ can be computed as

$$\text{Max:} \quad P(\boldsymbol{h}_j^{(l+1)} \leq q) = \prod_{i \in \mathcal{I}(j)} P(\boldsymbol{h}_i^{(l)} \leq q) \tag{87}$$

$$\text{Prob:} \quad P(\boldsymbol{h}_j^{(l+1)} = q) = \sum_{i \in \mathcal{I}(j)} \theta_i P(\boldsymbol{h}_i^{(l)} = q) \tag{88}$$

where $P(\boldsymbol{h}_i^{(l)} \leq q)$ is the *culminative mass function* of $P$. Complexities for both layers are $O(ID)$.

**Average Pooling and Depth-wise Convolutional Layer**  Both layers require additions of a medium number of inputs. We prove a *convolution theorem* for discrete random variables and show that *discrete Fourier transform (DFT)* (with *fast Fourier transform (FFT)*) can accelerate the additive computation. We also derive its backpropagation rule for compatibility of gradient-based learning.

**Theorem D.1 (Fast summation via discrete Fourier transform).** *Suppose $\boldsymbol{u}_i$ take values in $\{b_i, b_i + 1, \ldots, B_i\}$ between integers $b_i$ and $B_i$, then the summation $\boldsymbol{v} = \sum_{i=1}^{E} \boldsymbol{u}_i$ takes values between $b$ and $B$, where $b = \sum_{i=1}^{E} b_i$ and $B = \sum_{i=1}^{E} B_i$. Let $C^{\boldsymbol{v}}$, $C^{\boldsymbol{u}_i}$ be the discrete Fourier transforms of $P^{\boldsymbol{v}}$, $P^{\boldsymbol{u}_i}$ respectively, i.e.*

$$C^{\boldsymbol{v}}(f) = \sum_{v=b}^{B} P^{\boldsymbol{v}}(v) \exp(-j2\pi(v - b)f/(B - b + 1)) \tag{89a}$$

$$C^{\boldsymbol{u}_i}(f) = \sum_{u_i=b_i}^{B_i} P^{\boldsymbol{u}_i}(u_i) \exp(-j2\pi(u_i - b_i)f/(B_i - b_i + 1)) \tag{89b}$$

*Then $C^{\boldsymbol{v}}(f)$ is the element-wise product of all Fourier transforms $C^{\boldsymbol{u}_i}(f)$, i.e.*

$$C^{\boldsymbol{v}}(f) = \prod_{i=1}^{E} C^{\boldsymbol{u}_i}(f), \forall f \tag{90}$$

*Proof.* We only prove the theorem for two discrete random variable, and the extension to multiple variables can be proved using induction. Now consider $u_1 \in [b_1, B_1]$, $u_2 \in [b_2, B_2]$ and their sum $v = u_1 + u_2 \in [b, B]$, where $b = b_1 + b_2$ and $B = B_1 + B_2$. Denote the probability vectors of $u_1$, $u_2$ and $v$ as $P_1 \in \triangle^{B_1 - b_1}$, $P_2 \in \triangle^{B_2 - b_2}$ and $P \in \triangle^{B-b}$ respectively, then the entries in $P$ are computed with $P_1$ and $P_2$ by standard convolution as follows:

$$P(v) = \sum_{u_1=b_1}^{B_1} P_1(u_1)P_2(v - u_1) = \sum_{u_2=b_2}^{B_2} P_1(v - u_2)P_2(u_2), \ \forall v \in \{b, \cdots, B\} \quad (91)$$

The relation above is usually denoted as $P = P_1 * P_2$, where $*$ is the symbol for convolution. Now define the *characteristic functions* $C$, $C_1$, and $C_2$ as the discrete Fourier transform (DFT) of the probability vectors $P$, $P_1$ and $P_2$ respectively:

$$C(f) = \sum_{v=b}^{B} P(v) \exp\left(-j\frac{2\pi}{R}(v - b)f\right), \ f \in [R] \quad (92a)$$

$$C_i(f) = \sum_{u_i=b_i}^{B_i} P_i(u_i) \exp\left(-j\frac{2\pi}{R}(u_i - b_i)f\right), f \in [R] \quad (92b)$$

where $R$ controls the *resolution* of the Fourier transform (typically chosen as $R = B - b + 1$, i.e. the range of possible values). In this case, the characteristic functions are complex vectors of same length $R$, i.e. $C, C_1, C_2 \in \mathbb{C}^R$, and we denote the (functional) mappings as $C = \mathcal{F}(P)$ and $C_i = \mathcal{F}_i(P_i)$. Given a characteristic function, its original probability vector can be recovered by *inverse discrete Fourier transform (IDFT)*:

$$P(v) = \frac{1}{R} \sum_{f=0}^{R-1} C(f) \exp\left(j\frac{2\pi}{R}(v - b)f\right), \ \forall v \in \{b, \cdots, B\} \quad (93a)$$

$$P_i(u_i) = \frac{1}{R} \sum_{f=0}^{R-1} C_i(f) \exp\left(j\frac{2\pi}{R}(u_i - b_i)f\right), \ \forall u_i \in \{b_i, \cdots, B_i\} \quad (93b)$$

which we denote the inverse mapping as $P = \mathcal{F}^{-1}(C)$ and $P_i = \mathcal{F}_i^{-1}(C_i)$. Now we plug the convolution in Equation (91) into the characteristic function $C(f)$ in (92a) and rearrange accordingly:

$$C(f) = \sum_{v=b}^{B} \left(\sum_{u_1=b_1}^{B_1} P_1(u_1)P_2(v - u_1)\right) \exp\left(-j\frac{2\pi}{R}(v - b)f\right) \quad (94)$$

$$(\text{Let } u_2 = v - u_1) = \sum_{u_1=b_1}^{B_1} \sum_{u_2=b_2}^{B_2} P_1(u_1)P_2(u_2) \exp\left(-j\frac{2\pi}{R}(u_1 + u_2 - b)f\right) \quad (95)$$

$$(\text{Since } b = b_1 + b_2) = \left[\sum_{u_1=b_1}^{B_1} P_1(u_1) \exp\left(-j\frac{2\pi}{R}(u_1 - b_1)f\right)\right] \\ \left[\sum_{u_2=b_2}^{B_2} P_2(u_2) \exp\left(-j\frac{2\pi}{R}(u_2 - b_2)f\right)\right] \quad (96)$$

$$= C_1(f) \cdot C_2(f) \quad (97)$$

The equation above can therefore be written as $C = C_1 \circ C_2$, where we use $\circ$ to denote element-wise product. Thus, we have shown summation of discrete random variables corresponds to element-wise product of their characteristic functions. $\square$

With the theorem, addition of $E$ discrete random variables can be computed efficiently as follows

$$P^v = P^{u_1} * P^{u_2} * \cdots * P^{u_E} \quad (98)$$

$$= \mathcal{F}^{-1}\left(\mathcal{F}(P^{u_1}) \circ \mathcal{F}(P^{u_2}) \circ \cdots \circ \mathcal{F}(P^{u_E})\right) \quad (99)$$

where $\mathcal{F}$ denotes the Fourier transforms in Equations (93a) and (93b). If FFT is used in computing all DFT, the computational complexity of Equation (99) is $O(ER \log R) = O(E^2 D \log(ED))$ (since $R = O(ED)$), compared to $O(D^E)$ with direct tensor contraction.

**Backpropagation** When fast Fourier transform is used to accelerate additions in Bayesian quantized network, we need to derive the corresponding backpropagation rule, i.e. equations that relate $\partial \mathcal{L}/\partial P$ to $\{\partial \mathcal{L}/\partial P_i\}_{i=1}^I$. For this purpose, we break the computation in Equation (99) into three steps, and compute the derivative for each of these steps.

$$C_i = \mathcal{F}_i(P_i) \Longrightarrow \frac{\partial \mathcal{L}}{\partial P_i} = R \cdot \mathcal{F}_i^{-1}\left(\frac{\partial \mathcal{L}}{\partial C_i}\right) \tag{100a}$$

$$C = C_1 \circ \cdots \circ C_I \Longrightarrow \frac{\partial \mathcal{L}}{\partial C_i} = \frac{C}{C_i} \circ \frac{\partial \mathcal{L}}{\partial C} \tag{100b}$$

$$P = \mathcal{F}^{-1}(C) \Longrightarrow \frac{\partial \mathcal{L}}{\partial C} = R^{-1} \cdot \mathcal{F}\left(\frac{\partial \mathcal{L}}{\partial P}\right) \tag{100c}$$

where in (100b) we use $C/C_i$ to denote element-wise division. Since $P_i$ lies into real domain, we need to project the gradients back to real number in (100c). Putting all steps together:

$$\frac{\partial \mathcal{L}}{\partial P_i} = \Re\left\{\mathcal{F}_i^{-1}\left(\frac{C}{C_i} \circ \mathcal{F}\left(\frac{\partial \mathcal{L}}{\partial P}\right)\right)\right\}, \forall i \in [I] \tag{101}$$

### D.3 Large Fan-in Layers: Lyapunov Central Limit Approximation

In this part, we show that *Lyapunov central limit approximation (Lyapunov CLT)* accelerates probabilistic propagation in linear layers. For simplicity, we consider *fully-connected layer* in the derivations, but the results can be easily extended to types of *convolutional layers*. We conclude this part by deriving the corresponding backpropagation rules for the algorithm.

**Linear Layers** Linear layers (followed by a nonlinear transformations $\sigma(\cdot)$) are the most important building blocks in neural networks, which include *fully-connected* and *convolutional layers*. A linear layer is parameterized by a set of vectors $\boldsymbol{\theta}^{(l)}$'s, and maps $\boldsymbol{h}^{(l)} \in \mathbb{R}^I$ to $\boldsymbol{h}^{(l+1)} \in \mathbb{R}^J$ as

$$\boldsymbol{h}_j^{(l+1)} = \sigma\left(\sum_{i \in \mathcal{I}(j)} \boldsymbol{\theta}_{ji}^{(l)} \cdot \boldsymbol{h}_i^{(l)}\right) = \sigma\left(\sum_{i \in \mathcal{I}(j)} \boldsymbol{u}_{ji}^{(l)}\right) = \sigma\left(\boldsymbol{v}_j^{(l+1)}\right) \tag{102}$$

where $\boldsymbol{u}_{ji}^{(l)} = \boldsymbol{\theta}_{ji}^{(l)} \cdot \boldsymbol{h}_i^{(l)}$ and $\boldsymbol{v}_j^{(l+1)} = \sum_{i \in \mathcal{I}(j)} \boldsymbol{u}_{ji}^{(l)}$. The key difficulty here is to compute the distribution of $\boldsymbol{v}_j^{(l+1)}$ from the ones of $\{\boldsymbol{u}_{ji}^{(l)}\}_{i=1}^I$, i.e. *addition of a large number of random variables*.

**Theorem D.2 (Fast summation via Lyapunov Central Limit Theorem).** *Let $\boldsymbol{v}_j = \sigma(\tilde{\boldsymbol{v}}_j) = \sigma(\sum_{i \in \mathcal{I}(j)} \boldsymbol{\theta}_{ji} \boldsymbol{u}_i)$ be an activation of a linear layer followed by nonlinearity $\sigma$. Suppose both inputs $\{\boldsymbol{u}_i\}_{i \in \mathcal{I}}$ and parameters $\{\boldsymbol{\theta}_{ji}\}_{i \in \mathcal{I}(j)}$ have bounded variance, then for sufficiently large $|\mathcal{I}(j)|$, the distribution of $\tilde{\boldsymbol{v}}_j$ converges to a Gaussian distribution $\mathcal{N}(\tilde{\mu}_j, \tilde{\nu}_j)$ with mean and variance as*

$$\tilde{\mu}_j = \sum_{i=1}^I m_{ji}\mu_i \tag{103a}$$

$$\tilde{\nu}_j = \sum_{i=1}^I m_{ji}^2 \nu_i + v_{ji}\mu_i^2 + v_{ji}\nu_i \tag{103b}$$

*where $m_{ji} = \mathbb{E}[\theta_{ji}]$, $v_{ji} = \mathbb{V}[\theta_{ji}]$ and $\mu_i = \mathbb{E}[\boldsymbol{u}_i]$, $\nu_i = \mathbb{V}[\boldsymbol{u}_i]$. And if the nonlinear transform $\sigma$ is a sign function, each activation $\boldsymbol{v}_j$ follows a Bernoulli distribution $P(\boldsymbol{v}_j = 1) = \Phi(\tilde{\mu}_j/\sqrt{\tilde{\nu}_j})$, where $\Phi$ is the culminative probability function of a standard Gaussian distribution $\mathcal{N}(0, 1)$.*

*Proof.* The proof follows directly from the definitions of mean and variance:

$$\tilde{\mu}_j = \mathbb{E}\left[\sum_{i=1}^I \boldsymbol{\theta}_{ji}\, \boldsymbol{h}_i\right] = \sum_{i=1}^I \mathbb{E}[\boldsymbol{\theta}_{ji}\, \boldsymbol{h}_i] \tag{104}$$

$$= \sum_{i=1}^I \mathbb{E}[\boldsymbol{\theta}_{ji}]\, \mathbb{E}[\boldsymbol{h}_i] = \sum_{i=1}^I m_{ji}\mu_i \tag{105}$$

$$\tilde{\nu}_j = \mathbb{V}\left[\sum_{i=1}^{I} \boldsymbol{\theta}_{ji}\, \boldsymbol{h}_i\right] = \sum_{i=1}^{I} \mathbb{V}\left[\boldsymbol{\theta}_{ji}\, \boldsymbol{h}_i\right] \tag{106}$$

$$= \sum_{i=1}^{I} \left(\mathbb{E}\left[\boldsymbol{\theta}_{ji}^2\right] \mathbb{E}\left[\boldsymbol{h}_i^2\right] - \mathbb{E}\left[\boldsymbol{\theta}_{ji}^2\right] \mathbb{E}\left[\boldsymbol{h}_i^2\right]\right) \tag{107}$$

$$= \sum_{i=1}^{I} \left[\left(m_{ji}^2 + v_i\right)\left(\mu_i^2 + \nu_{ji}\right) - m_{ji}^2 \mu_i^2\right] \tag{108}$$

$$= \sum_{i=1}^{I} \left(m_{ji}^2 \nu_i + v_{ji}\mu_i^2 + v_{ji}\nu_i\right) \tag{109}$$

For fully-connected layers, these two equations can be concisely written in matrix forms:

$$\tilde{\mu} = M\mu \tag{110a}$$
$$\tilde{\nu} = \left(M^{\circ 2}\right)\nu + V\left(\mu^{\circ 2} + \nu\right) \tag{110b}$$

where $M^{\circ 2}$ and $\mu^{\circ 2}$ are element-wise square of $M$ and $\mu$ respectively. $\qquad\square$

**Backpropagation** With matrix forms, the backpropagation rules that relate $\partial\mathcal{L}/\partial\tilde{\psi}^{(l+1)} = \{\partial\mathcal{L}/\partial\tilde{\mu}, \partial\mathcal{L}/\partial\tilde{\nu}\}$ to $\partial\mathcal{L}/\partial\phi^{(l)} = \{\partial\mathcal{L}/\partial M, \partial\mathcal{L}/\partial V\}$ and $\partial\mathcal{L}/\partial\psi^{(l)} = \{\partial\mathcal{L}/\partial\mu, \partial\mathcal{L}/\partial\nu\}$ can be derived with matrix calculus.

$$\frac{\partial\mathcal{L}}{\partial M} = \left(\frac{\partial\mathcal{L}}{\partial\tilde{\mu}}\right)\mu^\top + 2M \circ \left[\left(\frac{\partial\mathcal{L}}{\partial\tilde{\nu}}\right)\nu^\top\right] \tag{111a}$$

$$\frac{\partial\mathcal{L}}{\partial V} = \left(\frac{\partial\mathcal{L}}{\partial\tilde{\nu}}\right)\left(\mu^{\circ 2}\right)^\top \tag{111b}$$

$$\frac{\partial\mathcal{L}}{\partial\mu} = M^\top\left(\frac{\partial\mathcal{L}}{\partial\tilde{\mu}}\right) + 2\mu \circ \left[V^\top\left(\frac{\partial\mathcal{L}}{\partial\tilde{\nu}}\right)\right] \tag{111c}$$

$$\frac{\partial\mathcal{L}}{\partial\nu} = \left(M^{\circ 2}\right)^\top\left(\frac{\partial\mathcal{L}}{\partial\tilde{\nu}}\right) \tag{111d}$$

Notice that these equations do not take into account the fact that $V$ implicitly defined with $M$ (i.e. $v_{ji}$ is defined upon $m_{ji}$). Therefore, we adjust the backpropagation rule for the probabilities: denote $Q_{ji}(d) = Q(\boldsymbol{\theta}_{ji} = \mathbb{Q}(d); \phi_{ji}^{(l)})$, then the backpropagation rule can be written in matrix form as

$$\frac{\partial\mathcal{L}}{\partial Q(d)} = \left(\frac{\partial\mathcal{L}}{\partial M} + \frac{\partial\mathcal{L}}{\partial V} \cdot \frac{\partial V}{\partial M}\right)\frac{\partial M}{\partial Q(d)} + \frac{\partial\mathcal{L}}{\partial V} \cdot \frac{\partial\nu}{\partial Q(d)} \tag{112}$$

$$= \mathbb{Q}(d) \cdot \frac{\partial\mathcal{L}}{\partial M} + 2(\mathbb{Q}(d) - M) \circ \left(\frac{\partial\mathcal{L}}{\partial V}\right) \tag{113}$$

Lastly, we derive the backpropagation rules for sign activations. Let $p_j$ denote the probability that the hidden unit $\boldsymbol{v}_j$ is activated as $p_j = \mathbf{Pr}[\boldsymbol{v}_j = 1|x]$, $\partial\mathcal{L}/p_j$ relates to $\{\partial\mathcal{L}/\partial\tilde{\mu}_j, \partial\mathcal{L}/\partial\tilde{\nu}_j\}$ as:

$$\frac{\partial p_j}{\partial\tilde{\mu}_j} = \mathcal{N}\left(\frac{\tilde{\mu}_j}{\sqrt{\tilde{\nu}_j}}\right) \tag{114a}$$

$$\frac{\partial p_j}{\partial\tilde{\nu}_j} = -\frac{1}{2\tilde{\nu}_j^{3/2}} \cdot \mathcal{N}\left(\frac{\tilde{\mu}_j}{\sqrt{\tilde{\nu}_j}}\right) \tag{114b}$$

## E  SUPPLEMENTARY MATERIAL FOR EXPERIMENTS

### E.1  NETWORK ARCHIECTURES

**(1)** For MNIST, Fashion-MNIST and KMNIST, we evaluate our models on both MLP and CNN. For MLP, we use a 3-layers network with 512 units in the first layer and 256 units in the second; and

for CNN, we use a 4-layers network with two $5 \times 5$ convolutional layers with $64$ channels followed by $2 \times 2$ average pooling, and two fully-connected layers with $1024$ hidden units. **(2)** For CIFAR10, we evaluate our models on a smaller version of VGG (Peters & Welling, 2018), which consists of 6 convolutional layers and 2 fully-connected layers: 2 x 128C3 – MP2 – 2 x 256C3 – MP2 – 2 x 512C3 – MP2 – 1024FC – SM10.

## E.2   MORE RESULTS FOR MULTI-LAYER PERCEPTRON (MLP)

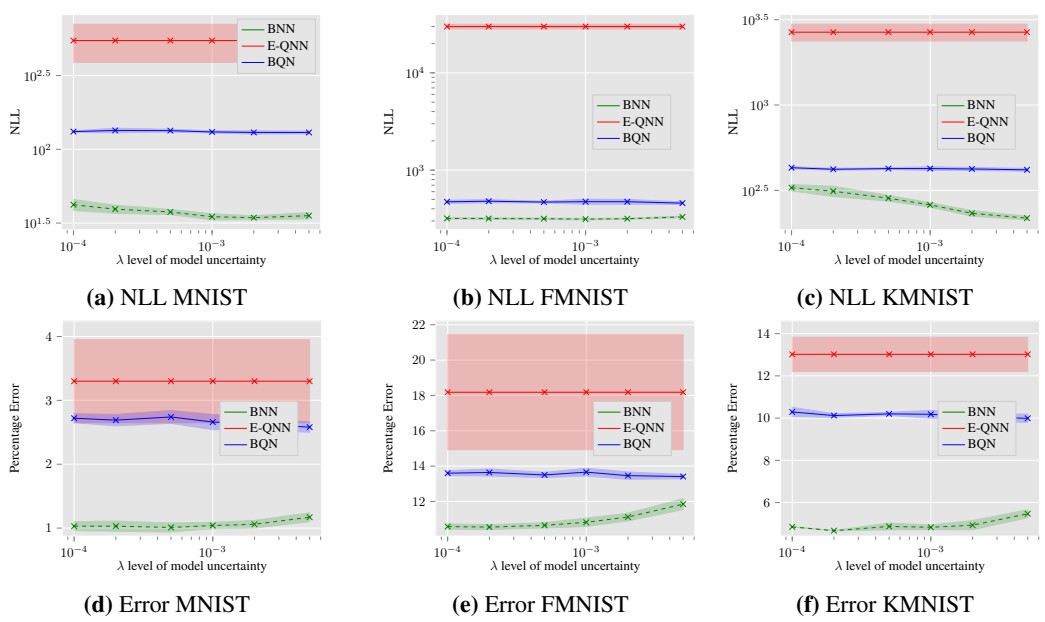

**Figure 4:** Comparison of the predictive performance of our BQNs against the E-QNN as well as the non-quantized BNN trained by SGVB on a MLP. Negative log-likelihood (NLL) which accounts for uncertainty and 0-1 test error which doesn't account for uncertainty are displayed.

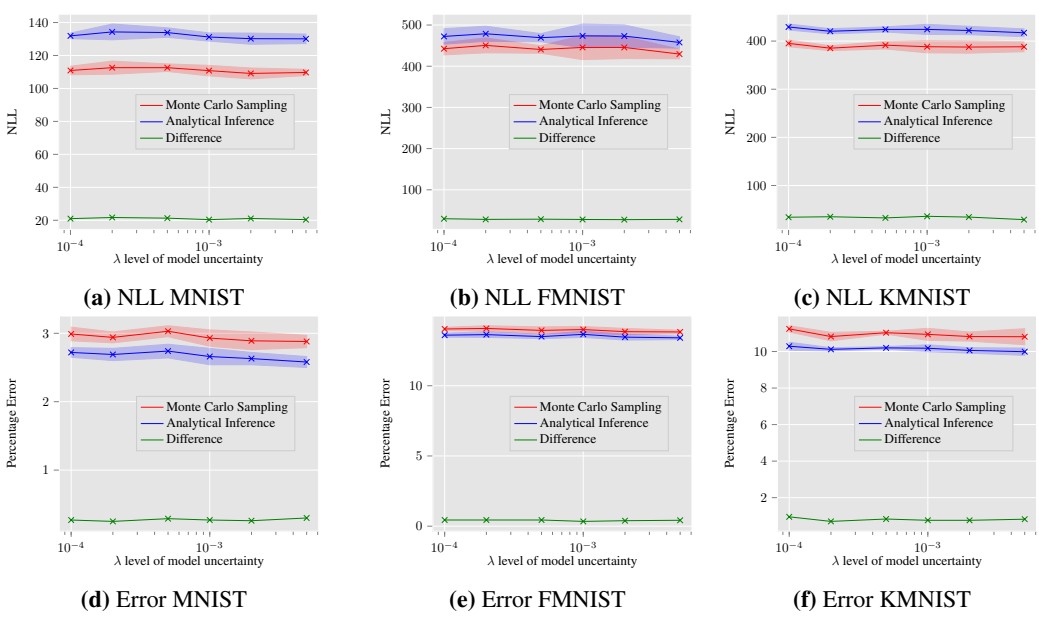

**Figure 5:** Illustration of mean-field approximation and tightness of alternative ELBO on a MLP. The performance gap between our analytical inference and the Monte Carlo Sampling is displayed.

### E.3  REGRESSION ON BOSTON HOUSING DATASET

In this part, we evaluate our proposed BQN on Boston housing dataset, a regression benchmark widely used in testing Bayesian neural networks (Hernández-Lobato & Adams, 2015; Ghosh et al., 2016) and Probabilistic neural networks (Wang et al., 2016). The dataset consists of $456$ training and $50$ test samples, each sample has $13$ features as input and a scalar (housing) price as output. Following Hernández-Lobato & Adams (2015); Ghosh et al. (2016); Wang et al. (2016), we train a two-layers network with $50$ hidden units, and report the performance in terms of *root mean square error (RMSE)* in Table 4. The results show that our BQN achieves lower RMSE compared to other models trained in a probabilistic/Bayesian way.

| Dataset | BQN | PBP (Ghosh et al., 2016) | EBP (Soudry et al., 2014) | NPN (Wang et al., 2016) |
|---|---|---|---|---|
| Boston | $2.04 \pm 0.07$ | $2.79 \pm 0.16$ | $3.14 \pm 0.93$ | $2.57 \pm$ NA |

**Table 4:** Performance of different networks in terms of RMSE. The numbers for BQN are averages over $10$ runs with different seeds, the standard deviation are exhibited following the $\pm$ sign. The results for PBP, EBP are from Ghosh et al. (2016), and the one for NPN is from (Wang et al., 2016).

