# OpenReview forum: "Sampling-Free Learning of Bayesian Quantized Neural Networks"
_ICLR.cc/2020/Conference — Accept (Poster)_

### Official Review · AnonReviewer3 · 2019-10-17
**Official Blind Review #3**

**Rating:** 6

**Review:**

(1) Contributions:
The paper proposes an algorithm for training quantized Bayesian neural networks (BQN). The method optimizes a lower bound to the ELBO that can be analytically computed for BQNs.

The paper shows that propagation in BQNs is a sequence of tensor contractions, for which there exist fast and efficient approximate algorithms.

BQNs show good performance on set of image classification benchmarks (MNIST, KMNIST, Fashion MNIST).

(2) Originality:
I am not aware of any prior works that attempts to do inference in quantized networks. The idea of BQNs and propagation as tensor contractions is interesting and novel. It is a cool result that in quantized networks, one can exactly calculate the distribution of the activations (h^(l)) and therefore get an accurate estimate of the ELBO.

It is also interesting that this algorithm can be implemented as a series of tensor contractions. This leads to a fast and efficient algorithm for training.

Regarding the novelty in the lower bound to the ELBO, variational Bayes makes the same approximation by sampling the layer activations (h^(1) .. h^(L-1)). In VB, when we sample the layer activations, we approximate the logarithm of the expectation by the logarithm of a Monte Carlo sample, which is a lower bound due to Jensen’s inequality. The only difference here is that this approximation is only done in the final hidden layer instead of every layer.

(2) Clarity:
I want to point out some issues with the writing.

Overall, I believe that the number of equations and theorems hinders the readability of the paper. It is very confusing to try and navigate the various notations and assess the significance of each result. I would recommend including a few key equations and theorems in the paper and leaving the rest to the appendix. The ideas and concepts in the paper could be easily communicated in writing.

I do not think this paper should be 9.5 pages. The work could be comfortably explained in the standard 8 pages. Namely, in Section 3.1 and 3.2, it is unnecessary to provide analytic gradients for the expressions (in the age of automatic differentiation). Section 4.3 should be left to the appendix. I found this section very confusing and it does not seem to be important in the narrative.

(4) Significance:
I think this work is significant, although I felt that the experiments section could have included more regression benchmarks as well as more difficult datasets such as CIFAR10.

(5) Impact:
This work would certainly be interesting to researchers and anyone working on quantized neural networks.

Overall assessment:
I think this paper presents interesting and novel ideas, but it has shortcomings in the presentation.

**Experience Assessment:**

I have published one or two papers in this area.

**Review Assessment: Checking Correctness Of Derivations And Theory:**

I did not assess the derivations or theory.

**Review Assessment: Checking Correctness Of Experiments:**

I assessed the sensibility of the experiments.

**Review Assessment: Thoroughness In Paper Reading:**

I read the paper at least twice and used my best judgement in assessing the paper.

---

> ### Author Response · Authors · 2019-11-15
> **Response to Official Blind Review #3**
>
> Thank you for your efforts in reviewing our submission.
>
> (1) Concerns of the presentation.
> In the revised version, we modified the presentation as suggested and reduced the length of the paper to 8 pages, moving technical details to the appendix so that the main body focuses on the ideas and concepts. Specifically, we removed the analytic gradients in the theorem statements, and we moved the equations in Section 4.3 to Appendix D.
>
> (2) On the novelty of our alternative ELBO.
> Our alternative ELBO provides a general approach to train Bayesian neural networks, which is not only applicable to BQN. Our approach can be preferable over traditional SGVB for two reasons: (a) In learning, it allows for a sampling-free algorithm to train Bayesian neural networks, which reduces the variance in gradient estimation caused by sampling. (b) In prediction, the trained model does not need to draw multiple samples to approximate the predictive distribution, thus reduces the computational expense.
> We have a paragraph in Appendix A.1 to discuss the differences between these two approaches in technical details.
>
> As discussed in Appendix A.1, the alternative ELBO has an additional advantage over the traditional SGVB when it applies to BQN. Since both weights and activations are quantized in BQN, the ELBO is not differentiable w.r.t. the drawn samples and as a result SGVB is not directly compatible to backpropagation. Our approach bypasses such difficulty by avoiding sampling from the weights or activations. (c) Therefore, the alternative ELBO can be applied to a broader scope of Bayesian neural networks than SGVB.
>
> (3) Regression benchmark and more difficult classification dataset such as CIFAR10.
> We have scaled up our BQNs for CIFAR10. The results are included in Section 5 (Table 1 and Figure 1). Our approach achieves both better negative log-likelihood (NLL) and 0-1 loss compared to the baseline ensemble QNN (E-QNN). Even when compared with the state-of-the-art Bayesian-CNN [a], our BQN impressively achieves better NLL, with a small increase in 0-1 loss (which is expected since Bayesian-CNN considers continuous-valued weights).  As we discussed in the paper, the more reasonable performance measure should be NLL for Bayesian models.
>
> We have also tested our BQNs for regression tasks on the Boston Housing dataset. The results are in Appendix E.3. Our BQN outperforms a set of baselines as we achieve a lower RMSE compared to other neural networks trained in a probabilistic/Bayesian way.
>
> [a] Shridhar, Kumar, Felix Laumann, and Marcus Liwicki. "A comprehensive guide to bayesian convolutional neural network with variational inference." arXiv preprint arXiv:1901.02731 (2019).

---

### Official Review · AnonReviewer1 · 2019-10-22
**Official Blind Review #1**

**Rating:** 6

**Review:**

This paper proposed Bayesian Quantized Neural Networks (BQN), which is a rough Bayesian treatment for quantized neural networks (QNN). The advantages the BQN over QNN is that it provides better uncertainty estimates as well as accuracy. The disadvantage is that BQN loses the memory efficient property of QNN, that is, BQN in general still needs to store real-value parameters, which parameterize the binary weights. The authors compare BQN to EQNN and show that BQN outperforms the latter.

In general, the paper is well organized and fairly readable, though the first 2 sections could have been more concise. I appreciate the careful description of notations.

The topic of Bayesian formulation and inference in the context of QNN is interesting. I like the analysis provided for handling small/medium/large fan-in for Bayesian QNN.

One of my concerns is that some description in the provided theorems (as well as the proof) is not very rigorous and sometimes misleading. For example, Theorem 3.2 claims that there is an analytic form for Bayesian inference in softmax layers. One would expect that it is either an exact integration over the distribution of h, or a valid bound. Unfortunately, it is neither. Looking at the proof, it seems it is a rather coarse approximation. Note that in the Appendix, inequality in Equation (42) and Equation (47) takes different direction, meaning that this is not a valid bound for the error term L. The authors may want to adjust the description to make it more rigorous and avoid confusion.

The paper describes BQN in very general terms. It is unclear what distributions are used to parameterize the weights in the experiments. Do you use Bernoulli distributions to parameterize the binary weights {-1, 1}, and learn the real-valued parameters for the distributions during training?

Theorem 3.1 is straightforward. The rough approximation provided in Theorem 3.2 is interesting. It would be interesting to compare this with Gaussian natural-parameter networks, which bypass the softmax problem using multi-logistic regression for classification, where the convolution of Gaussian and sigmoid functions has a closed form.

In terms of related work, references such as natural-parameter networks (NPN) and its variants (one of which specifically handles the softmax case) are missing [1, 2]. Note that, similarly to BQN, NPN also provides a general probabilistic (Bayesian) framework for sampling-free learning/prediction. Theorem 3.3 is also covered in Section 3.2 of [1].

This brings me to another of my concerns. The only baseline provided is E-QNN from three years ago. There are various state-of-the-art baselines, either on QNN or BNN. It would certainly make the paper much stronger if such baselines are included, especially when the authors decide to take up the full ten-page space in this submission.


Minor:

P2: bf Bayesian -> Bayesian

P2: theta that predicted -> theta that predict

P6: Lemma 4.2 -> Theorem 4.2

P6: by with -> with

Assuming most of my concerns can be addressed during the feedback period, I tend to give a ‘weak accept’ to this submission.

Natural-Parameter Networks: A Class of Probabilistic Neural Networks, NIPS 2016
Feed-forward Propagation in Probabilistic Neural Networks with Categorical and Max Layers, ICLR 2018

**Experience Assessment:**

I have published in this field for several years.

**Review Assessment: Checking Correctness Of Derivations And Theory:**

I assessed the sensibility of the derivations and theory.

**Review Assessment: Checking Correctness Of Experiments:**

I carefully checked the experiments.

**Review Assessment: Thoroughness In Paper Reading:**

I read the paper thoroughly.

---

> ### Author Response · Authors · 2019-11-15
> **Response to Official Blind Review #1**
>
> Thank you very much for your detailed comments and questions. We respond to them one by one:
>
> (1) Concerns about Theorem 3.2 not being rigorous.
> There were two typos in our original submission: (a) The \leq sign in Eq.(43) (Eq.(47) in the original version) should have been a \geq, since -log() is a concave function; (b) the equal sign in Theorem 3.2 should have been a \geq, i.e. the analytic form is a further lower bound of the alternative ELBO in Theorem 3.1. Therefore the analytic form in Theorem 3.2 is a strict bound of the original ELBO, not an approximation.
>
> (2) Question of what distributions are used to parameterize the weights in the experiments.
> In our experiments we consider binary neural networks where both weights and activations are quantized to {-1, 1}. Therefore, we use Bernoulli distribution to model the binary weights. As discussed at the beginning of Section 4.2, the distribution parameters are stored in log-space to avoid projected gradient ascent in optimization.
>
> (3) Comparison of Theorem 3.2 with Gaussian natural-parameter networks, which bypass the softmax problem using multi-logistic regression for classification.
> We assume you are referring to Equations (5) and (6) in [1], where the convolution of a Gaussian distribution with a sigmoid function is approximated by another sigmoid. The results therein are not directly applicable to our approach, since the alternative ELBO in Theorem 3.1 requires us to compute the convolution between log-probability (not the probability itself) and the nonlinear output function. It is not clear whether an analytic lower bound can be obtained using multi-logistic regression. However, as you have commented, the results are useful to compute the predictive distribution in multi-logistic regression. In fact, we compute a similar approximation in Appendix C.3, where the convolution between Gaussian and softmax is approximated by another softmax (see Equations (60)-(67) in the revised paper for details).
>
> (4) Comparison of our BQN with natural-parameter networks (NPN).
> Thank you for pointing out these references. We have added them to our revision. Natural-parameter networks are probabilistic neural networks where both activations and weights are modeled using distributions in the exponential family. Our work is closely related to NPN in two ways: (a) Our BQN can be considered as a special case of NPN since the Bernoulli/categorical distribution belongs to the exponential family. (b) While the original NPN [1] is not trained in a principled Bayesian way (the negated loss function defined between the predicted distribution and the targets is not a strict lower bound), our analysis in Section 3 provides a principled way to train Bayesian NPNs.
>
> (5) Analysis of argmax and softmax layers in NPN.
> A variant of Gaussian NPN [2] proposes to use argmax as the output layer, and shows that it can be approximated by softmax function in NPNs. It will be interesting to analyze argmax layer using Section 3 in our work. In future work, we will derive analytic forms for other output layers, such as argmax layer as mentioned and layers for other tasks (basides classification and regression).
>
> (6) Concerns about baselines used in our paper.
> The BNN baseline [a] used in the experiments section is the state-of-the-art, to the best of our knowledge, which scales the SGVB method for large convolutional neural networks. The approach to train binary neural network in our paper, to the best of our knowledge, is also the state-of-the-art among methods that don’t require a pre-trained continuous-valued neural network as the initialization.
>
> As far as we are aware, the state-of-the-art strategy for training binary neural networks is the probabilistic method in [b], which requires initialization from a pre-trained continuous-valued network as shown in the ablation studies of [b] (see Section 4.6 therein). Since we only consider methods that train from scratch using random initialization in our paper, we do not use the method in [b] as our baseline. Also, we are not able to find their source code to reproduce their results. We will investigate proper initialization of BQNs in a future work.
>
> Finally, thank you for your detailed typographical corrections. We have fixed the typos you noted in our revision, and we have reduced the paper to 8 pages as suggested.
>
> [a] Shridhar, Kumar, Felix Laumann, and Marcus Liwicki. "A comprehensive guide to bayesian convolutional neural network with variational inference." arXiv preprint arXiv:1901.02731 (2019).
>
> [b] Peters, Jorn WT, and Max Welling. "Probabilistic binary neural networks." arXiv preprint arXiv:1809.03368 (2018).
>
> [1] Natural-Parameter Networks: A Class of Probabilistic Neural Networks, NIPS 2016
>
> [2] Feed-forward Propagation in Probabilistic Neural Networks with Categorical and Max Layers, ICLR 2018

---

### Official Review · AnonReviewer2 · 2019-10-23
**Official Blind Review #2**

**Rating:** 6

**Review:**

This work proposes a set of efficient algorithms for learning and prediction in Bayesian quantized networks, which allows for differentiable learning without the need to sampling. From this point of view, this work targets at solving a real and interesting problem in BNN. I am not an expert in this area, and I can only comment on broad picture.

Some questions:
-	The proposed method is claimed to work on large-scale datasets. However, all datasets selected in experiment are black and white images with low resolution (28 x 28 ?). A ‘real’ large-scale dataset is expected. The current experiment results are far from sufficient.
-	The competitive methods are limited. If the author claims BNN in general can have well-calibrated uncertainties, non-BNN methods should also be considered. Further, when the data has heavy tail or is not Gaussian-distributed, will the model still give well calibrated results?
-	Does this proposed sampling free BQN have faster computational speed than standard BNN? If yes, it is also desired to have a comparison.
-	I am totally lost at section 4.3. It seems to give a fast way of computing Eq.(12). A general introduction at the beginning or the end of section 4.3 could be really helpful. Further, I’m not clear why the approximation made in the model is good enough. A pseudo-algorithm is desired for clarification.


**Experience Assessment:**

I have read many papers in this area.

**Review Assessment: Checking Correctness Of Derivations And Theory:**

I assessed the sensibility of the derivations and theory.

**Review Assessment: Checking Correctness Of Experiments:**

I assessed the sensibility of the experiments.

**Review Assessment: Thoroughness In Paper Reading:**

I read the paper at least twice and used my best judgement in assessing the paper.

---

> ### Author Response · Authors · 2019-11-15
> **Response to Official Blind Review #2**
>
> Thank you very much for your thoughtful comments. In response to your questions:
>
> (1) Concern about the scale of the experiments.
> We have added experiments on CIFAR10 to our paper. The results are updated in Section 5 (see Table 1 and Figure 1). The results show that our method has higher accuracy and better-calibrated uncertainty compared to the ensemble of QNNs (E-QNN) trained with straight-through estimator. Our BQNs also achieve better NLL compared to state-of-the-art continuous-valued Bayesian CNN. As we only evaluate BQNs with parameter values in {-1, 1} in this paper, a small degradation in predictive accuracy compared to Bayesian CNNs is expected.
>
> (2) Question about comparing against non-BNN methods.
> The ensemble of quantized neural networks (E-QNN) to which we compare our method is a non-BNN method. The results in Table 1 and Figure 1 show that our proposed BQN outperforms the non-Bayesian E-QNN baseline.
>
> (3) Question about whether BQN can be applied to non-Gaussian data.
> Like all other machine learning models, our BQN does not have any assumption on data distribution, therefore can be applied to data with arbitrary distribution.
>
> (4) Computational efficiency of sampling-free BQN over standard BNN.
> It is hard to directly compare the efficiency of BQN and BNN, since our BQN is sampling-free, and standard BNNs require sampling to produce predictions. While a single forward pass in BQN is more expensive than a single sample of BNN (specifically, 50% more expensive if local-reparameterization trick is used for sampling), multiple samples must be drawn from a standard BNN to make a prediction. Therefore, the question is: how many samples need to be drawn for BNN to produce an estimate with as low variance as BQN? This will depend on the learning problem and the network architecture.
>
> (5) Readability of Section 4.3.
> We have rewritten Section 4.3 for a more concise illustration of our fast algorithms. As for the question of how good the approximation we have made is, Figure 2 shows the results of experiments designed to address this question. Figure 2 shows that the gap between probabilistic propagation with approximation and Monte-Carlo evaluation without approximation is small for all scenarios, which justifies the approximations we use in BQNs.
>
> Again, thank you very much for your time and effort in reviewing our submission.

---

### Author Response · Authors · 2019-11-15
**Revised the paper as suggested by the reviewers**

We thank all reviewers for the valuable suggestions to improve our work. We have updated the following parts of the paper:

(1) We reorganize the writing and shrink the paper to 8 pages as suggested by the reviewers. (a) We move the discussion of why the alternative ELBO in Theorem 3.1 is compatible to backpropagation to Appendix C.1. (b) We move the analytic form of the alternative ELBO for regression to Appendix C.3, and we remove the derivatives in Theorem 3.2 as suggested by Reviewer 3. (c) We rewrite Section 4.3 to improve readability. We add a paragraph as a general introduction to the algorithms, and we also move the technical equations of the algorithms to Appendix D.

(2) We add experiments on CIFAR10 for larger-scale classification, as shown in Section 5 (Table 1 and Figure 1). Our approach achieves both better negative log-likelihood (NLL) and 0-1 loss compared to the baseline ensemble QNN (E-QNN). Even when compared with the state-of-the-art Bayesian-CNN, our BQN impressively achieves lower NLL, with a small increase in 0-1 loss (which is expected since Bayesian-CNN considers continuous-valued weights).  As we discussed in the paper, the more reasonable performance measure should be NLL for Bayesian models.

(3) We add experiments on Boston Housing dataset for regression task, as shown in Appendix E.3. Our BQN outperforms a set of baselines as we achieve a lower RMSE compared to other neural networks trained in a probabilistic/Bayesian way.

(4) We fix typos in Theorem 3.2. In the original version, Theorem 3.2 stated that the alternative ELBO is equal to the analytic form, which was a typo. In this revised version, we correct it to be an inequality. i.e. the analytic form is another lower bound of the alternative ELBO in Theorem 3.1, and therefore also a strict lower bound of the original ELBO. In the proof of Theorem 3.2 (Appendix C.2) there was a typo when using Jensen’s inequality, where the inequality should be \geq instead of \leq.

(5) We add references to natural-parameter networks (NPN) [a, b]. As pointed out by Reviewer 3, our work is closely related to NPN. Indeed, our BQN can be interpreted as a categorical NPN, but trained with principled variational inference. We add a paragraph in Appendix A.1 (related works) to discuss the relationship between BQN and NPN.

[a] Wang, Hao, S. H. I. Xingjian, and Dit-Yan Yeung. "Natural-parameter networks: A class of probabilistic neural networks." Advances in Neural Information Processing Systems. 2016.

[b] Shekhovtsov, Alexander, and Boris Flach. "Feed-forward Propagation in Probabilistic Neural Networks with Categorical and Max Layers." (2018).

---

### Decision · Program_Chairs · 2019-12-19

**Decision:**

Accept (Poster)

**Comment:**

This paper proposes Bayesian quantized networks and efficient algorithms for learning and prediction of these networks. The reviewers generally thought that this was a novel and interesting paper.  There were a few concerns about the clarity of parts of the paper and the experimental results. These concerns were addressed during the discussion phase, and the reviewers agree that the paper should be accepted.